# DAZL mediates a broad translational program regulating expansion and differentiation of spermatogonial progenitors

Maria M Mikedis[1], Yuting Fan[1,2], Peter K Nicholls[1], Tsutomu Endo[1], Emily K Jackson[1,3], Sarah A Cobb[1], Dirk G de Rooij[1], David C Page[1,3,4]*

[1]Whitehead Institute, Cambridge, United States; [2]Reproductive Medicine Center, Sixth Affiliated Hospital, Sun Yat-sen University, Guangzhou, China; [3]Department of Biology, Massachusetts Institute of Technology, Cambridge, United States; [4]Howard Hughes Medical Institute, Whitehead Institute, Cambridge, United States

**Abstract** Fertility across metazoa requires the germline-specific DAZ family of RNA-binding proteins. Here we examine whether DAZL directly regulates progenitor spermatogonia using a conditional genetic mouse model and in vivo biochemical approaches combined with chemical synchronization of spermatogenesis. We find that the absence of *Dazl* impairs both expansion and differentiation of the spermatogonial progenitor population. In undifferentiated spermatogonia, DAZL binds the 3' UTRs of ~2,500 protein-coding genes. Some targets are known regulators of spermatogonial proliferation and differentiation while others are broadly expressed, dosage-sensitive factors that control transcription and RNA metabolism. DAZL binds 3' UTR sites conserved across vertebrates at a UGUU(U/A) motif. By assessing ribosome occupancy in undifferentiated spermatogonia, we find that DAZL increases translation of its targets. In total, DAZL orchestrates a broad translational program that amplifies protein levels of key spermatogonial and gene regulatory factors to promote the expansion and differentiation of progenitor spermatogonia.

*For correspondence: dcpage@wi.mit.edu

**Competing interests:** The authors declare that no competing interests exist.

## Introduction

The germline-specific DAZ family of RNA-binding proteins functions in germ cell development across metazoa. This family is comprised of Y-linked DAZ and its autosomal homologs DAZL (DAZ-like) and BOULE, all of which contain a highly conserved RNA recognition motif (RRM) and at least one DAZ repeat. BOULE is widely conserved across metazoa from sea anemones through humans, while DAZL is limited to vertebrates, and DAZ is further limited to Old World monkeys and apes (*Saxena et al., 1996*; *Xu et al., 2001*). In humans, deletions encompassing the Y chromosome's Azoospermia Factor C (*AZFc*) region, which contains all four copies of the *DAZ* gene, are among the most common known genetic causes of spermatogenic failure, accounting for 10% of cases of azoospermia (no sperm detected in semen) or severe oligozoospermia (abnormally low number of sperm detected in semen) in the absence of any physical obstruction (*Ambulkar et al., 2015*; *Fu et al., 2012*; *Girardi et al., 1997*; *Mascarenhas et al., 2016*; *Nakahori et al., 1996*; *Reijo et al., 1995*; *Simoni et al., 1997*; *Vogt et al., 1996*). However, the mechanistic basis for the DAZ family's role in spermatogenesis remains poorly defined.

*Dazl* is expressed in the embryonic germ line as well as during adult oogenesis and spermatogenesis (*Seligman and Page, 1998*) and therefore likely functions at multiple stages of germline development. In mice, genetic loss of *Dazl* causes infertility in both sexes (*Ruggiu et al., 1997*). DAZL is

first required during embryogenesis for germ cell determination (*Gill et al., 2011*; *Lin and Page, 2005*; *Nicholls et al., 2019b*). On mixed genetic backgrounds and the inbred 129 strain, a small number of *Dazl*-null germ cells survive into adulthood with defects in spermatogonial differentiation ($A_{al}$ to $A_1$ transition) (*Schrans-Stassen et al., 2001*) and in meiosis I (*Saunders et al., 2003*). However, because these studies examined a complete knockout of *Dazl*, it is unclear whether these adult phenotypes are a primary defect due to loss of DAZL activity in the adult germ line or a secondary effect of the absence of DAZL during embryogenesis. To address this, one study conditionally deleted *Dazl* after its embryonic requirement and reported roles for DAZL in spermatogonial stem cell maintenance, meiosis I, and spermatid development (*Li et al., 2019*).

DAZL interacts with the 3' UTRs of factors that contribute to multiple stages of spermatogenesis, but it is not clear what transcripts DAZL targets in specific spermatogenic cell types because these targets have been identified in either cultured primordial germ cell-like cells (*Chen et al., 2014*), ovaries (*Rosario et al., 2017*), or whole testes containing many stages of spermatogenesis (*Chen et al., 2014*; *Li et al., 2019*; *Zagore et al., 2018*). DAZL functions as a translational enhancer within the oocyte (*Collier et al., 2005*; *Reynolds et al., 2005*; *Reynolds et al., 2007*; *Sousa Martins et al., 2016*). However, during spermatogenesis, and particularly in spermatogonia, there is conflicting evidence whether DAZL regulates translation (*Li et al., 2019*; *Reynolds et al., 2005*; *Reynolds et al., 2007*) or RNA stability (*Zagore et al., 2018*).

Here, we study mouse *Dazl* using genetic and biochemical tools to define a function for the DAZ family in spermatogenesis. Using a conditional mouse model, we find that DAZL promotes expansion and differentiation of progenitor spermatogonia. By combining chemical synchronization of spermatogenesis with iCLIP and translational profiling, we demonstrate that DAZL mediates a broad translational program spanning ~2,500 protein-coding genes, including spermatogonial and gene regulatory factors, in undifferentiated spermatogonia in vivo. We propose that these novel insights into how DAZL functions within undifferentiated spermatogonia illuminate DAZ's role in human spermatogenesis.

## Results

### DAZL promotes expansion and differentiation of the spermatogonial progenitor population

To determine when DAZL could play a direct role in spermatogenesis, we defined DAZL protein expression in the postnatal male germ line. At birth, male germ cells are present as mitotically quiescent gonocytes (also known as prospermatogonia). At postnatal days (P) 0 and 4, we found that gonocytes robustly expressed *Dazl*, as marked by a *Dazl*:tdTomato reporter (*Figure 1—figure supplement 1A*).

Shortly after birth, gonocytes resume proliferation and mature into spermatogonia, including spermatogonial stem cells. These spermatogonia initiate spermatogenesis, which encompasses a series of transit-amplifying mitotic divisions, followed by meiosis and then the cellular differentiation of spermiogenesis to form spermatozoa. DAZL protein expression has been previously reported in multiple spermatogenic cell types, including spermatogonia, spermatocytes, and round spermatids (*Li et al., 2019*; *Ruggiu et al., 1997*; *Xu et al., 2001*). Here, we sought to clarify DAZL's expression patterns via immunohistochemistry. DAZL protein was robustly expressed in type A, Intermediate, and type B spermatogonia as well as in spermatocytes in meiotic prophase I (*Figure 1—figure supplement 1B–C*). However, DAZL was not detected in secondary spermatocytes in meiotic prophase II, or in round or elongating spermatids, after the completion of meiosis. These data provide greater detail as to the spermatogenic cell types that express DAZL (*Li et al., 2019*; *Ruggiu et al., 1997*; *Xu et al., 2001*) and are at odds with a previous study reporting DAZL expression in round spermatids (*Li et al., 2019*), potentially reflecting differences in antibody specificity or sensitivity of detection.

Next, we investigated whether DAZL is required for spermatogonial development, independent of DAZL's earlier requirement in germ cell determination (~E10.5-12.5). Using a conditional *Dazl* allele with germline-specific Cre recombinase $Ddx4^{Cre}$, which is active from ~E14.5 onward (*Figure 1A*), we generated *Dazl* conditional knockout mice ($Dazl^{2L/-}$; $Ddx4^{Cre/+}$; referred to as *Dazl* cKO) alongside phenotypically wild-type control animals ($Dazl^{2L/+}$; $Ddx4^{Cre/+}$). Germ cells persist in

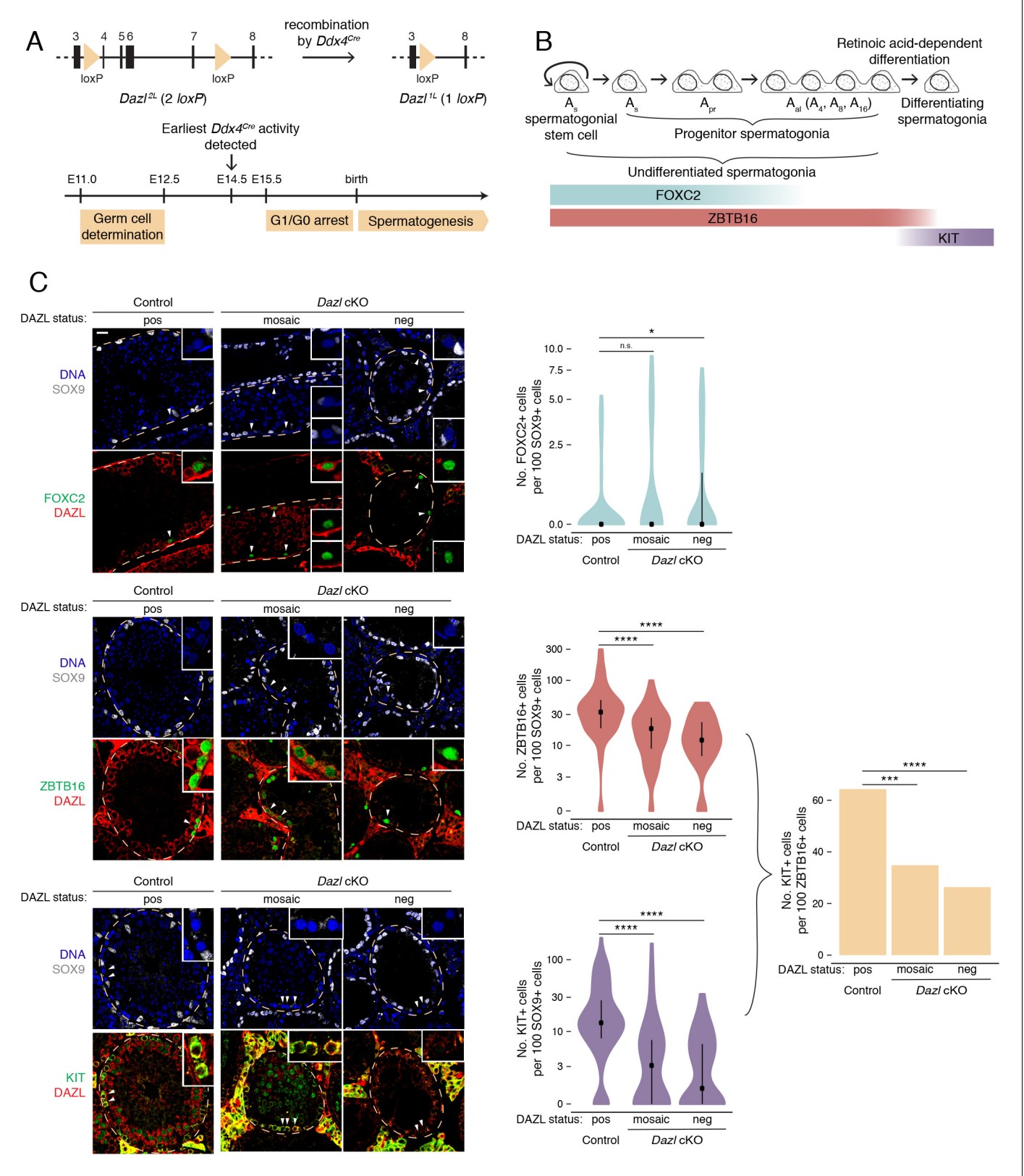

**Figure 1.** DAZL promotes spermatogonial expansion and differentiation. (**A**) Schematic of *Dazl²ᴸ* (two *loxP*) and *Dazl¹ᴸ* (one *loxP*) alleles. Recombination of *Dazl²ᴸ* allele via *Ddx4ᶜʳᵉ* allele yields *Dazl¹ᴸ* allele in germ cells. (**B**) Schematic of spermatogenesis from Asingle (As) spermatogonial stem cells to progenitor spermatogonia to differentiating spermatogonia, with expression of spermatogonial markers. As spermatogonial stem cells

*Figure 1 continued on next page*

*Figure 1 continued*

self-renew while also giving rise to $A_s$ progenitor spermatogonia, which differentiate without self-renewal. Subsequent divisions with incomplete cytokinesis produce chains of two ($A_{paired}$, $A_{pr}$) as well as chains of four, eight, and sixteen ($A_{aligned}$, $A_{al}$) progenitor spermatogonia. Progenitors form differentiating spermatogonia in response to retinoic acid. In addition, chains of $A_{pr}$ and $A_{al}$ progenitors can fragment to form $A_s$ spermatogonial stem cells, particularly under conditions of stress (not shown). (C) Quantification of FOXC2, ZBTB16, and KIT-positive spermatogonia (green) in histological sections of control and *Dazl* cKO adult testes. SOX9 marks Sertoli cells (gray), DAZL is in red, and DAPI marks DNA (blue). Select tubules are outlined via dotted orange line. Cells enlarged within insets are highlighted by arrowheads. Insets show spermatogonia that are DAZL-positive and FOXC2-positive (control and mosaic cKO panels); DAZL-negative and FOXC2-positive (mosaic and DAZL-negative cKO panels); DAZL-positive and ZBTB16-positive (control and mosaic cKO panels); DAZL-negative and ZBTB16-positive (DAZL-negative cKO panel); DAZL-positive and KIT-positive (control and mosaic cKO panels); and DAZL-negative and KIT-negative (DAZL-negative cKO panel). Scale bar = 20 μm. Populations were quantified from 50 tubules per animal, with two animals per genotype. Violin plots display medians with interquartile ranges. Difference between normalized FOXC2, ZBTB16, and KIT-positive populations was statistically assessed via two-sided Mann-Whitney U test. Difference between normalized KIT-positive populations over normalized ZBTB16-positive populations was statistically assessed via two-sided odds ratio. *, p<0.05; ***, p<0.001; ****, p<0.0001.

The online version of this article includes the following source data and figure supplement(s) for figure 1:

**Source data 1.** RNA-seq analysis of *Pou5f1*:EGFP-positive spermatogonia from control and *Dazl* cKO testes.
**Source data 2.** Quantification of spermatogonial subpopulations and tubule cross-section area in control and *Dazl* cKO testes.
**Figure supplement 1.** DAZL expression in postnatal gonocytes and during spermatogenesis.
**Figure supplement 2.** Conditional deletion of *Dazl* in spermatogonia.
**Figure supplement 3.** Characterization of *Pou5f1*:EGFP-positive spermatogonia in *Dazl* cKO.

adult *Dazl* cKO testes, but these animals are infertile (*Nicholls et al., 2019b*). However, conditional deletion of *Dazl* in the adult was incomplete, and while some tubules lacked DAZL protein, other tubules continued to express DAZL (*Nicholls et al., 2019b*). Here, at P10, we found that while many tubules contained DAZL-negative germ cells, nearly 100% of *Dazl* cKO tubule cross sections contained at least one DAZL-positive cell (*Figure 1—figure supplement 2A–B*). However, at 6 months, only ~40% contained at least one DAZL-positive cell, compared with 100% of controls (*Figure 1— figure supplement 2B*). We therefore focused our analysis on aged animals, in which the conditional deletion of *Dazl* was more complete.

First, we asked whether spermatogonial stem cells and early progenitor ($A_s$ and $A_{pr}$) spermatogonia, as identified by FOXC2 (*Wei et al., 2018*; *Figure 1B*), are maintained in the absence of DAZL at 6 months. Using immunofluorescent staining, we quantified this stem/progenitor population in seminiferous tubule cross sections (*Figure 1C*). To account for shrinkage of the seminiferous tubules and concomitant changes in cell distribution caused by disrupted spermatogenesis (*Figure 1—figure supplement 2C*), the FOXC2 population was normalized to the population of SOX9-positive Sertoli cells, which lack DAZL and do not proliferate in adult testes (*Kluin et al., 1984*; *Vergouwen et al., 1991*). To account for the incomplete conditional deletion, *Dazl* cKO tubules that lacked DAZL-positive germ cells ('DAZL-negative') and those that exhibited one or more DAZL-positive germ cell ('mosaic') were analyzed separately. In DAZL-negative tubules from *Dazl* cKO testes, the FOXC2-positive population of stem/progenitor spermatogonia was modestly larger than in controls (two-sided Mann-Whitney U test, p=0.019), but no such difference was observed between mosaic tubules from *Dazl* cKO testes and controls (two-sided Mann-Whitney U test, p=0.095). We conclude that, within our cKO model, loss of *Dazl* has minimal impact on stem/progenitor spermatogonia within the testes.

Next, we examined whether the broader population of undifferentiated spermatogonia (stem cells as well as early and late progenitors), as identified by ZBTB16 (also known as PLZF) (*Figure 1B*), is affected in the *Dazl* cKO. The median number of ZBTB16-positive cells was reduced by 40–60% in cKO tubules (two-sided Mann-Whitney U test, $p=1.85\times10^{-6}$ and $p=1.57\times10^{-10}$ for mosaic and DAZL-negative cKO tubules, respectively; *Figure 1C*). Given that the numbers of stem/progenitor spermatogonia are not reduced, we conclude that fewer late progenitor ($A_{al}$) spermatogonia form in the absence of *Dazl*.

We then asked if the progenitor spermatogonia that do form are able to differentiate by quantifying the KIT-positive population of differentiating spermatogonia (*Figure 1B*). The median number of KIT-positive cells was reduced by 70–90% in cKO tubules (two-sided Mann-Whitney U test, $p=7.06\times10^{-9}$ and $p=1.33\times10^{-10}$ for mosaic and DAZL-negative cKO tubules, respectively; *Figure 1C*). We calculated the total rate of spermatogonial differentiation as the ratio of the

normalized number of KIT-positive cells over that of ZBTB16-positive cells. In control testes, the differentiation rate was ~65%, which is consistent with previously observed rates (reviewed in *de Rooij, 2017*; *Tegelenbosch and de Rooij, 1993*). By contrast, the rate of spermatogonial differentiation was reduced to 25–35% in cKO testes (two-sided odds ratio, p=$1.06 \times 10^{-4}$ and p=$1.40 \times 10^{-6}$ for mosaic and DAZL-negative cKO tubules, respectively; *Figure 1C*). Therefore, without *Dazl*, progenitor spermatogonia do not differentiate efficiently.

To confirm these results, we isolated late progenitor and early differentiating spermatogonia expressing the *Pou5f1*:EGFP reporter at 6–8 months via FACS (*Figure 1—figure supplement 3A–B*; *Garcia and Hofmann, 2012*; *La et al., 2018b*) and analyzed gene expression via RNA-seq (*Figure 1—figure supplement 3C–D*). Several genes associated with stem/progenitor spermatogonia exhibited increased expression in the *Pou5f1*:EGFP-positive spermatogonial population from the *Dazl* cKO compared with those from the control, while several genes expressed in progenitor spermatogonia or the broader population of undifferentiated spermatogonia exhibited reduced abundance (adjusted p<0.05; *Figure 1—figure supplement 3D*). However, *Zbtb16* and *Kit* RNA levels appeared unaffected, presumably because RNA-seq analysis of the bulk *Pou5f1*:EGFP-positive population does not provide the single-cell resolution of our immunohistological quantification (*Figure 1C*). It is also possible that the reduced population of progenitor and differentiating spermatogonia in the *Dazl* cKO expressed higher levels of *Zbtb16* and *Kit* to compensate for the loss of DAZL. Regardless, other markers in this population-level RNA-seq analysis corroborate our immunohistological analysis.

In sum, the loss of *Dazl* results in dramatically fewer differentiating spermatogonia for two reasons: progenitor spermatogonia fail to fully expand their population, and they do not differentiate efficiently.

## DAZL broadly targets the transcriptome in progenitor spermatogonia

Given that DAZL promotes the expansion and differentiation of progenitor spermatogonia, we set out to identify DAZL's targets in undifferentiated spermatogonia using iCLIP (individual nucleotide resolution in vivo UV crosslinking and immunoprecipitation; *Huppertz et al., 2014*). To obtain the large numbers of undifferentiated spermatogonia needed for this biochemical analysis, we developmentally synchronized spermatogenesis (*Hogarth et al., 2013*; *Romer et al., 2018*; *Figure 2A*). By chemically regulating the level of retinoic acid, which is required for spermatogonial differentiation (*Endo et al., 2015*; *van Pelt and de Rooij, 1990*), we synchronized spermatogonial development and greatly enriched for undifferentiated spermatogonia in the testes. The successful accumulation of undifferentiated spermatogonia, without contamination from later stages of spermatogenesis, was verified in a testis biopsy by both histological analysis and immunohistochemical staining for STRA8 protein (*Figure 2—figure supplement 1A*), which is expressed at spermatogonial differentiation (*Endo et al., 2015*). To further verify synchronization, we isolated the synchronized germline population from *ROSA26^{tdTomato/+}*; *Ddx4^{Cre/+}* testes (synchronize, stage, and sort or '3S' protocol). *Ddx4^{Cre}* activates the *ROSA26^{tdTomato}* reporter in the vast majority (~95%) of germ cells by P1 (*Nicholls et al., 2019b*), prior to the initiation of synchronization treatments. We analyzed the 3S undifferentiated spermatogonia by RNA-seq (*Figure 2A*, *Figure 2—figure supplement 1B–D*). We compared these data to RNA-seq datasets of spermatogonia from unsynchronized testes (*Kubo et al., 2015*; *La et al., 2018b*; *Maezawa et al., 2017*). As expected, our 3S undifferentiated spermatogonia were more similar to sorted undifferentiated spermatogonia than sorted differentiating spermatogonia and exhibited minimal expression of differentiation markers *Kit* and *Stra8* (*Figure 2—figure supplement 1E–F*). These data demonstrate that chemical synchronization of spermatogenesis provides an enriched population of undifferentiated spermatogonia.

We carried out DAZL iCLIP on synchronized and staged ('2S') wild-type testes (*Figure 2A*; *Figure 2—figure supplement 2A–C*). Crosslinked nucleotides captured by iCLIP were used to identify peaks in three biological replicates (*Figure 2B*). Within each replicate, the vast majority of peaks (≥89%) were in the 3' UTR of mRNAs (*Figure 2B–C*, *Figure 2—figure supplement 2D*). We therefore defined DAZL's binding sites as 3' UTR peaks present in at least two of the three replicates (*Figure 2C*).

DAZL binding sites corresponded to 2,633 genes (*Figure 2C*), representing 19.5% of all protein-coding genes expressed in undifferentiated spermatogonia. Based on the transcript abundance of these genes, 31.8% of all mRNA molecules in undifferentiated spermatogonia contained DAZL

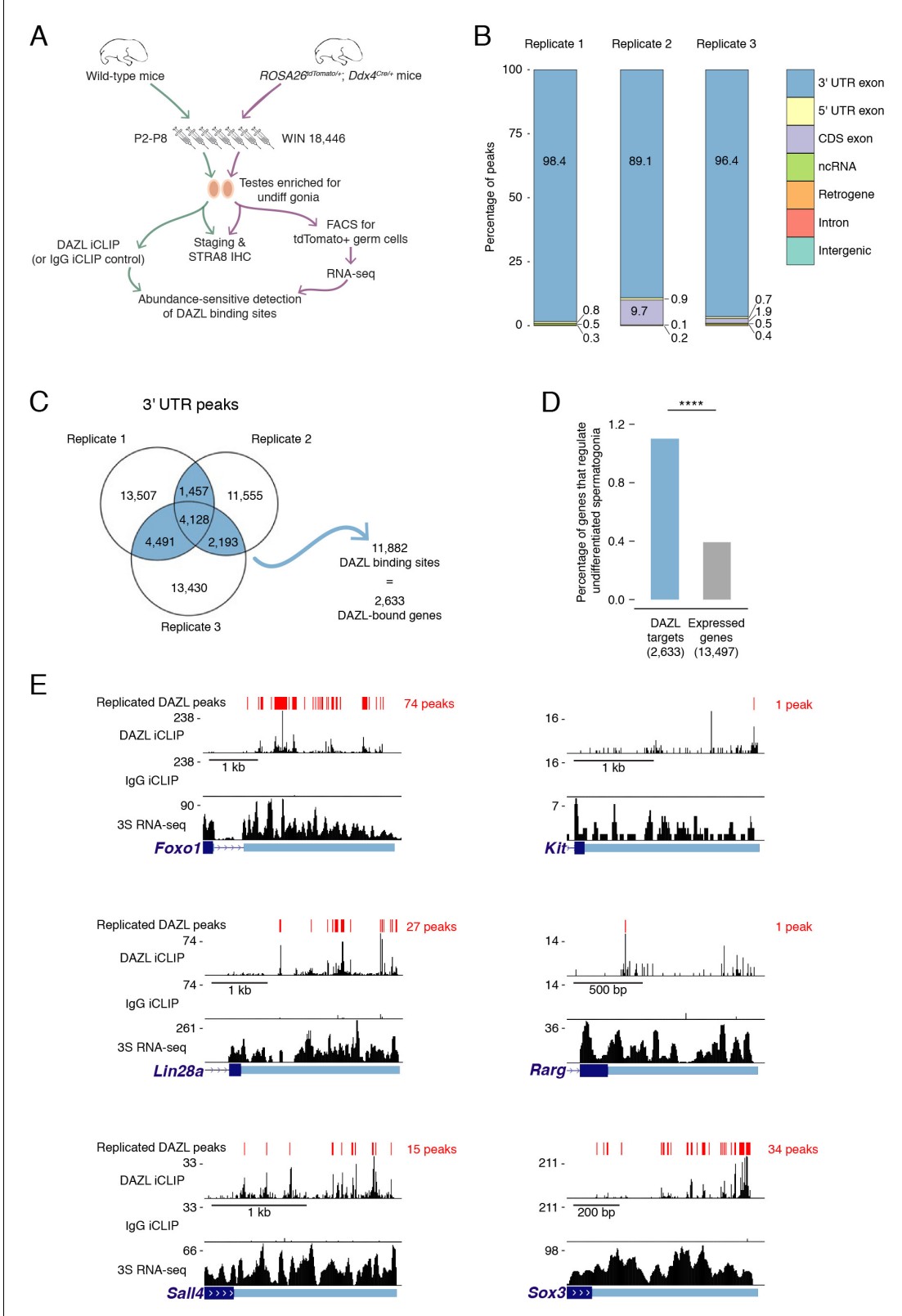

**Figure 2.** Identification of DAZL-bound transcripts in undifferentiated spermatogonia via iCLIP reveals DAZL's regulation of spermatogonial factors. (**A**) Schematic of the synchronization of spermatogenesis to obtain undifferentiated spermatogonia via the 2S method for iCLIP (green arrows) and via the 3S method for RNA-seq (purple arrows). WIN 18,446 was used to synchronize spermatogenesis by blocking spermatogonial differentiation, and samples were histologically staged to verify successful synchronization. For 3S samples, germ cells were sorted from synchronized testes. (**B**) Genomic

*Figure 2 continued on next page*

*Figure 2 continued*

distribution of DAZL iCLIP peaks identified in three biological replicates (TPM ≥1; FDR < 0.05). (C) Venn diagram showing overlap of DAZL iCLIP peaks in expressed 3′ UTRs (TPM ≥1) among three biological replicates. Replicated peaks (i.e., present in at least two of three replicates) were identified. After merging replicated peaks that fell on consecutive nucleotides, 11,882 DAZL binding sites (present in at least two of three replicates; highlighted in blue) were identified. These binding sites correspond to 2,633 genes, which are designated as the DAZL-bound genes. (D) Enrichment of factors that regulate the development and differentiation of undifferentiated spermatogonia in DAZL targets compared with all genes expressed in undifferentiated spermatogonia (one-tailed hypergeometric test). Number of genes (n) in each group designated in parentheses in labels along x-axis.****, p<0.0001. (E) DAZL iCLIP, IgG iCLIP, and 3S RNA-seq gene tracks showing exemplary DAZL-bound genes that are required for spermatogonial proliferation and expansion (*Lin28a and Sox3*) or differentiation (*Kit*, *Foxo1*, *Rarg*, and *Sall4*). Each iCLIP track represents the crosslinked sites from the sum of unique reads from three biological replicates. The RNA-seq track represents the sum of two biological replicates. The scale of each gene track is marked on the left. 3′ UTRs are in light blue.

The online version of this article includes the following source data and figure supplement(s) for figure 2:

**Source data 1.** Genes that regulate development and differentiation in undifferentiated spermatogonia.
**Source data 2.** Replicated DAZL iCLIP peaks and transcript expression levels in 3S undifferentiated spermatogonia.
**Source data 3.** Comparison of 3S undifferentiated spermatogonia to previously published spermatogonial datasets.
**Figure supplement 1.** Isolation of undifferentiated spermatogonia via the 2S (synchronization and staging) and 3S (synchronization, staging, and sorting) strategies.
**Figure supplement 2.** DAZL iCLIP.

binding sites. Therefore, DAZL interacts with about one-third of the transcriptome in undifferentiated spermatogonia. However, DAZL is not unusual in its number of targets, as CLIP experiments have revealed that many RNA-binding proteins interact with a comparably large number of targets (*Van Nostrand et al., 2020*; *Yamaji et al., 2017*).

Given that *Dazl* promotes expansion and differentiation of progenitor spermatogonia, we sought mechanistic insights into how DAZL instructs spermatogonial development. To identify DAZL targets that are known spermatogonial regulators, we queried a set of genes previously annotated as regulating development and differentiation of undifferentiated spermatogonia (*Figure 2—source data 1*) (reviewed in *Mecklenburg and Hermann, 2016*). While these genes comprised only 0.4% of expressed protein-coding genes in undifferentiated spermatogonia (53 of 13,497 expressed genes), they represented 1.1% of DAZL targets (29 genes of 2,633 DAZL targets), representing an enrichment of spermatogonial factors (*Figure 2D*; one-tailed hypergeometric test; p=1.71×10$^{-8}$). Two of these DAZL-targeted factors, *Lin28a* and *Sox3*, promote the formation and proliferation of progenitor spermatogonia (*Chakraborty et al., 2014*; *McAninch et al., 2020*) and may contribute to the diminished progenitor population observed in the *Dazl* cKO (*Figure 2E*). Additional factors, such as *Foxo1*, *Kit*, *Rarg*, and *Sall4*, promote spermatogonial differentiation (*de Rooij et al., 1999*; *Gely-Pernot et al., 2012*; *Goertz et al., 2011*; *Hobbs et al., 2012*; *Yoshinaga et al., 1991*), consistent with the reduced spermatogonial differentiation in the *Dazl* cKO (*Figure 2E*). Therefore, DAZL preferentially interacts with factors that regulate both expansion and differentiation of progenitor spermatogonia, consistent with its genetically demonstrated roles in these cells.

## DAZL interacts with broadly expressed, dosage-sensitive regulators of fundamental cellular processes

Since known spermatogonial factors comprise just a minor fraction of DAZL's targets (*Figure 2D*), we sought to characterize other factors with which DAZL interacts. Gene Ontology (GO) analysis of all DAZL targets revealed enrichment for regulators of general transcription and RNA splicing (*Figure 3A*). At the level of transcriptional regulation, DAZL binds to *Ep300* (also known as *p300*), a histone acetyltransferase; *Polr2d*, a component of RNA polymerase II; and *Taf4b*, a germline-enriched subunit of the general transcription factor TFIID (*Figure 3B*). At the level of splicing regulation, DAZL binds to *Celf1*, an RNA-binding protein involved in alternative splicing; *Sf1*, a splicing factor required for spliceosome assembly; and *Snrpb*, a core spliceosomal protein (*Figure 3B*). Via such interactions, DAZL-mediated regulation has the potential to broadly shape the transcriptome of progenitor spermatogonia.

While GO analysis highlighted that DAZL interacts with many factors that regulate fundamental cellular processes, it failed to reveal any enrichment for germline-specific processes. We therefore tested whether DAZL targets tend not to be germline-specific factors. Using RNA-seq data from 12

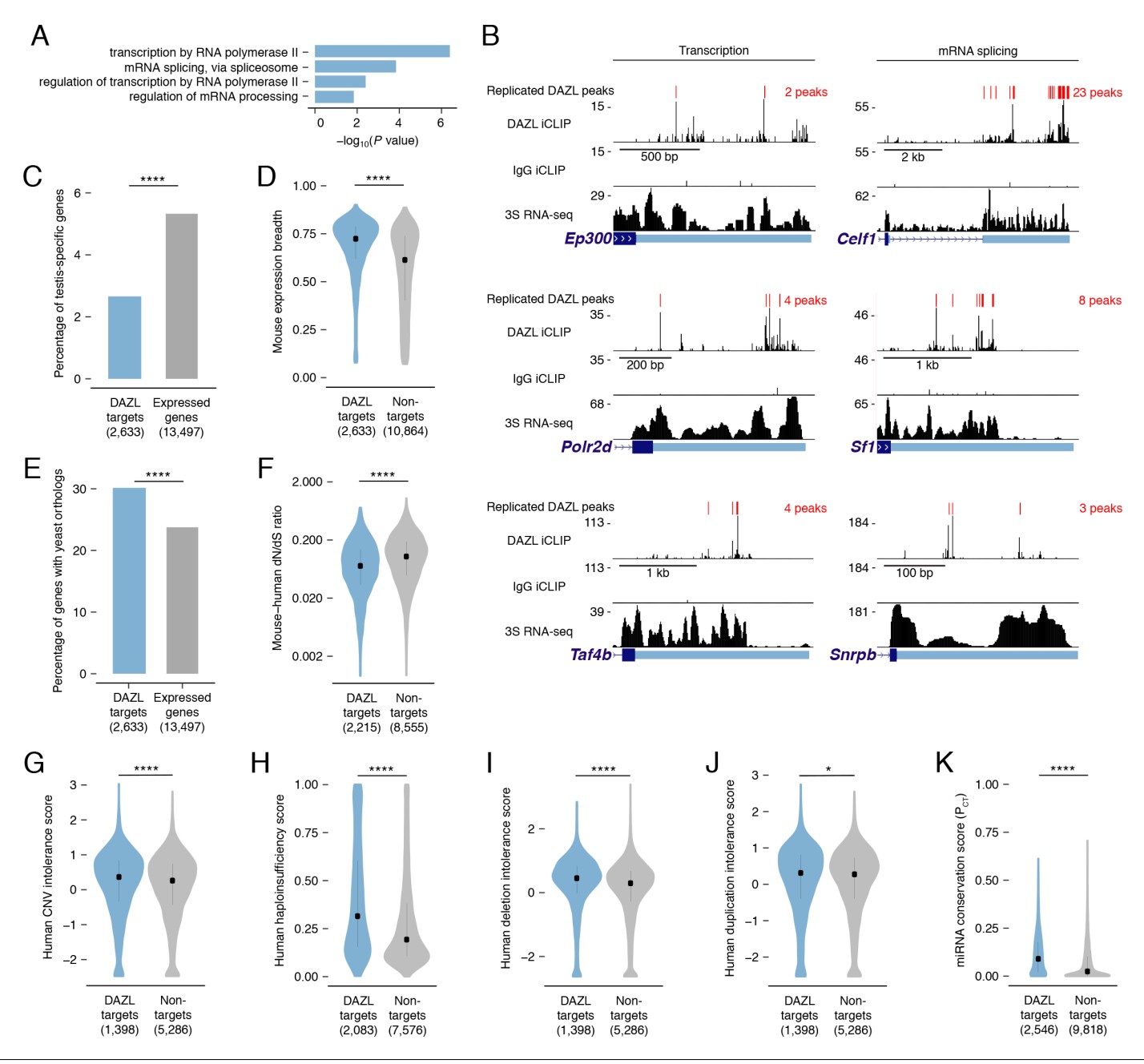

**Figure 3.** DAZL targets are enriched for broadly expressed, dosage-sensitive regulators of basic cellular processes. (**A**) All statistically enriched GO biological processes, excluding parental nodes, for 2,633 DAZL-targeted genes compared with all genes expressed in undifferentiated spermatogonia (TPM ≥1) using the PANTHER GO Slim annotation set. *P* values are from binomial tests with Bonferroni correction. (**B**) DAZL iCLIP, IgG iCLIP, and 3S RNA-seq gene tracks showing exemplary DAZL-bound genes that are involved in transcription and RNA splicing. Gene tracks as described in *Figure 2E*. (**C**) DAZL targets are depleted of testis-specific factors compared with all genes expressed in undifferentiated spermatogonia (one-tailed hypergeometric test). (**D**) DAZL targets are more broadly expressed than nontargets within 12 adult mouse tissues (one-sided Mann-Whitney U test). (**E**) DAZL targets are more likely to have a yeast ortholog (one-tailed hypergeometric test). (**F**) DAZL targets have a reduced ratio of nonsynonymous substitutions per nonsynonymous site to synonymous substitutions per synonymous site (dN/dS) in alignments with human orthologs (one-sided Mann-Whitney U test). (**G**) Human orthologs of DAZL's targets exhibit increased intolerance for copy number variation (CNV) (one-sided Mann-Whitney U test). (**H**) The human orthologs of DAZL's targets exhibit increased probability of haploinsufficiency (one-sided Mann-Whitney U test). (**I**) The human orthologs of DAZL's targets exhibit increased intolerance for deletions (one-sided Mann-Whitney U test). (**J**) The human orthologs of DAZL's targets exhibit increased intolerance for duplications (one-sided Mann-Whitney U test). (**K**) DAZL targets exhibit higher miRNA conservation scores ($P_{CT}$) (one-sided Mann-Whitney U test). Violin plots display medians with interquartile ranges; number of genes (n) in each group designated in parentheses in labels along x-axis. *, p<0.05; ****, p<0.0001.

*Figure 3 continued on next page*

Figure 3 continued

The online version of this article includes the following source data for figure 3:

**Source data 1.** GO analysis of DAZL targets.
**Source data 2.** Testis-specific genes and expression breadth for genes expressed in undifferentiated spermatogonia.
**Source data 3.** Mouse:yeast orthology status for genes expressed in undifferentiated spermatogonia.
**Source data 4.** dN/dS ratios of mouse:human orthologs for genes expressed in undifferentiated spermatogonia.
**Source data 5.** Haploinsufficiency scores, ExAC data, and mean miRNA $P_{CT}$ scores for genes expressed in undifferentiated spermatogonia.

male mouse tissues (*Naqvi et al., 2019*), we identified testis-specific genes (*Figure 3—source data 2*) and confirmed that *Dazl* expression is testis-specific, as previously described (*Nicholls et al., 2019b*). We found that testis-specific factors comprised 5.3% of the genes expressed in undifferentiated spermatogonia but only 2.7% of DAZL targets, representing a depletion of testis-specific factors among DAZL targets (one-tailed hypergeometric test; p=1.78×10$^{-13}$; *Figure 3C*). Given this, we assessed whether DAZL's targets were broadly expressed. Across 12 mouse tissues, DAZL targets exhibited greater expression breadth than non-target genes expressed in undifferentiated spermatogonia (one-sided Mann-Whitney U test, p<2.2×10$^{-16}$; *Figure 3D*). Therefore, while DAZL itself is a germ cell-specific factor, its targets are biased towards broadly expressed genes that are not unique to germ cells. The broad expression of DAZL targets is consistent with their regulation of fundamental cellular processes.

To further test whether factors bound by DAZL regulate fundamental cellular processes, we examined the conservation of DAZL targets. If these targets are critical to fundamental cellular functions, then they should be preferentially conserved across the 900–1,400 million years of evolution that separate mice from yeast (*Berbee et al., 2017*). This conservation would exist independent of DAZL family members, which only evolved in multicellular organisms and are absent from *Saccharomyces cerevisiae*. While 23.8% of mouse genes expressed in undifferentiated spermatogonia had orthologs in *Saccharomyces cerevisiae*, 30.2% of DAZL targets had an ortholog, representing an enrichment in yeast orthologs among DAZL's targets (one-tailed hypergeometric test; p=2.09×10$^{-17}$; *Figure 3E*). Therefore, DAZL preferentially interacts with factors conserved between mouse and yeast, indicating that these factors support basic cellular functions.

Given that DAZL preferentially interacts with conserved factors, we hypothesized that DAZL targets are subject to stronger purifying selection, which limits changes to the amino acid sequence and consequently causes an excess of synonymous substitutions relative to nonsynonymous ones. In mouse:human orthologs, DAZL targets exhibited a reduced ratio of nonsynonymous to synonymous substitution rates (dN/dS) compared with nontargets (one-sided Mann-Whitney U test, p<2.2×10$^{-16}$; *Figure 3F*). Therefore, DAZL targets are under strong purifying selection, consistent with their roles in fundamental cellular processes.

DAZL targets include conserved components of transcriptional and splicing complexes. To maintain stoichiometry, dosage of proteins within these complexes is strictly regulated (*Veitia and Birchler, 2009*; *Veitia and Potier, 2015*). Therefore, we asked whether DAZL targets are sensitive to dosage decreases and increases. First, as a metric of general dosage sensitivity, we assessed human copy number variation (CNV) intolerance scores (*Lek et al., 2016*). The human orthologs of genes targeted by DAZL were more intolerant of CNV than nontargets expressed in undifferentiated spermatogonia (one-sided Mann-Whitney U test, p=7.51×10$^{-5}$; *Figure 3G*). Next, we examined whether DAZL targets are sensitive to dosage decreases. Compared with nontargets, human orthologs of DAZL targets had higher probabilities of displaying haploinsufficiency (one-sided Mann-Whitney U test, p<2.2×10$^{-16}$; *Figure 3H*; *Huang et al., 2010*) and were more intolerant of deletions (one-sided Mann-Whitney U test, p=5.6×10$^{-16}$; *Figure 3I*; *Lek et al., 2016*). We then examined whether DAZL targets are similarly sensitive to dosage increases. Human orthologs of genes targeted by DAZL exhibited a greater intolerance for duplications than nontargets (one-sided Mann-Whitney U test, p=0.045; *Figure 3J*; *Lek et al., 2016*). We also examined conserved miRNA targeting, as genes that are sensitive to dosage increases exhibit conserved targeting by miRNAs, which modulate gene dosage by lowering the levels of their mRNA targets (*Bartel, 2009*; *Naqvi et al., 2018*). Indeed, DAZL targets had higher probabilities of conserved targeting ($P_{CT}$ scores) (*Agarwal et al., 2015*; *Friedman et al., 2009*), and were consequently more sensitive to dosage increases, than nontargets

(one-sided Mann-Whitney U test, p<$2.2\times10^{-16}$; *Figure 3K*). Therefore, DAZL targets are sensitive to both increases and decreases in dosage.

In summary, within undifferentiated spermatogonia, DAZL binds transcripts that facilitate proliferation and differentiation of progenitor spermatogonia, consistent with our genetic findings in the conditional model. However, DAZL also binds a broad set of conserved, dosage-sensitive transcripts that control the fundamental cellular processes of transcription and splicing, which may also affect the efficiency of proliferation and differentiation in spermatogonia.

## DAZL binds a UGUU(U/A) motif

To biochemically characterize DAZL's binding preferences, we analyzed DAZL's binding sites for de novo motif discovery. The top motif from two independent tools contained GUU (*Figure 4A*), consistent with previously identified motifs (*Chen et al., 2011*; *Jenkins et al., 2011*; *Li et al., 2019*; *Maegawa et al., 2002*; *Reynolds et al., 2005*; *Zagore et al., 2018*). We then asked whether DAZL-bound GUU motifs are flanked by a specific sequence. By comparing GUU motifs from DAZL binding sites to unbound GUUs from the same 3' UTRs, we expanded the DAZL-bound motif to UGUU(U/A) (*Figure 4B*). Beyond this five-nucleotide motif, there was a bias for U's 5' to the motif as well as U's or A's 3' to the motif. Many RNA binding proteins exhibit similarly broad flanking sequence preferences (*Dominguez et al., 2018*).

As iCLIP captured nucleotides that directly interact with DAZL, we verified that the UGUU(U/A) motif was overrepresented at crosslinked nucleotides, relative to the GUU and UUU identified by the de novo motif analysis (*Figure 4C*). To verify that all five nucleotides of the UGUU(U/A) motif contribute to DAZL's binding preferences, we examined truncations of the motif (i.e., UGUU and GUU(U/A)) and confirmed that they were less enriched than the full-length motif (*Figure 4—figure supplement 1A*).

The UGUU(U/A) motif is consistent with the UGUU and UUUGUUU motifs identified by previous studies (*Chen et al., 2014*; *Li et al., 2019*). Other studies had characterized DAZL's motif as GUUG (*Zagore et al., 2018*), GUUC (*Maegawa et al., 2002*; *Reynolds et al., 2005*), or UUU(C/G)UUU (*Chen et al., 2011*). All of these motifs share features with, but are distinct from, the UGUU(U/A) motif identified here. To test whether previously identified motifs similarly capture DAZL's binding preferences, we examined their enrichment at crosslinked nucleotides. We found that each motif exhibited less enrichment than UGUU(U/A) at crosslinked nucleotides (*Figure 4—figure supplement 1A*) and at DAZL binding sites (*Figure 4—figure supplement 1B*). In addition, UGUU(U/A) was more enriched than these other motifs in our reanalysis of an independent DAZL iCLIP dataset derived from P6 testes (*Figure 4—figure supplement 1C*; *Zagore et al., 2018*). Finally, we found that the UGUU(U/A) motif was enriched at the small number of DAZL iCLIP peaks identified outside of 3' UTRs (*Figure 4—figure supplement 1D*). In total, UGUU(U/A) robustly captures DAZL's binding preferences.

## DAZL's 3' UTR binding sites are conserved among vertebrates

Mouse DAZL, human DAZL, and human DAZ likely have similar binding preferences, as their RNA-binding domains are highly conserved (*Jenkins et al., 2011*; *Saxena et al., 1996*). To explore whether DAZL's binding sites in mice may be bound by DAZ family members during spermatogenesis in humans as well as in other vertebrates, we asked whether mouse DAZL binding sites are conserved among vertebrates. First, we analyzed phyloP scores, which quantify the conservation of each individual nucleotide (*Figure 4D*). We found that DAZL-crosslinked nucleotides were significantly more conserved than other 3' UTR nucleotides (two-sided Mann-Whitney U test, p<$2.2\times10^{-16}$). Next, we analyzed phastCons conservation scores, a per-nucleotide score that reflects the conservation of each nucleotide as well as its neighbors (*Figure 4D*). DAZL-crosslinked nucleotides were located within more conserved regions of the 3' UTR than noncrosslinked nucleotides (two-sided Mann-Whitney U test, p<$2.2\times10^{-16}$). These highly conserved sites include one of DAZL's 3' UTR binding sites in *Celf1* (*Figure 4E*), as well as binding sites in *Lin28a* and *Ep300* (*Figure 4—figure supplement 1E–F*).

Next, we characterized DAZL's binding behavior along the 3' UTR. The majority of DAZL targets had at least two binding sites (*Figure 4F*). When ranked according to the number of DAZL binding sites, the top 5% of DAZL targets had ≥15 sites and included spermatogonial factors *Foxo1*, *Lin28a*,

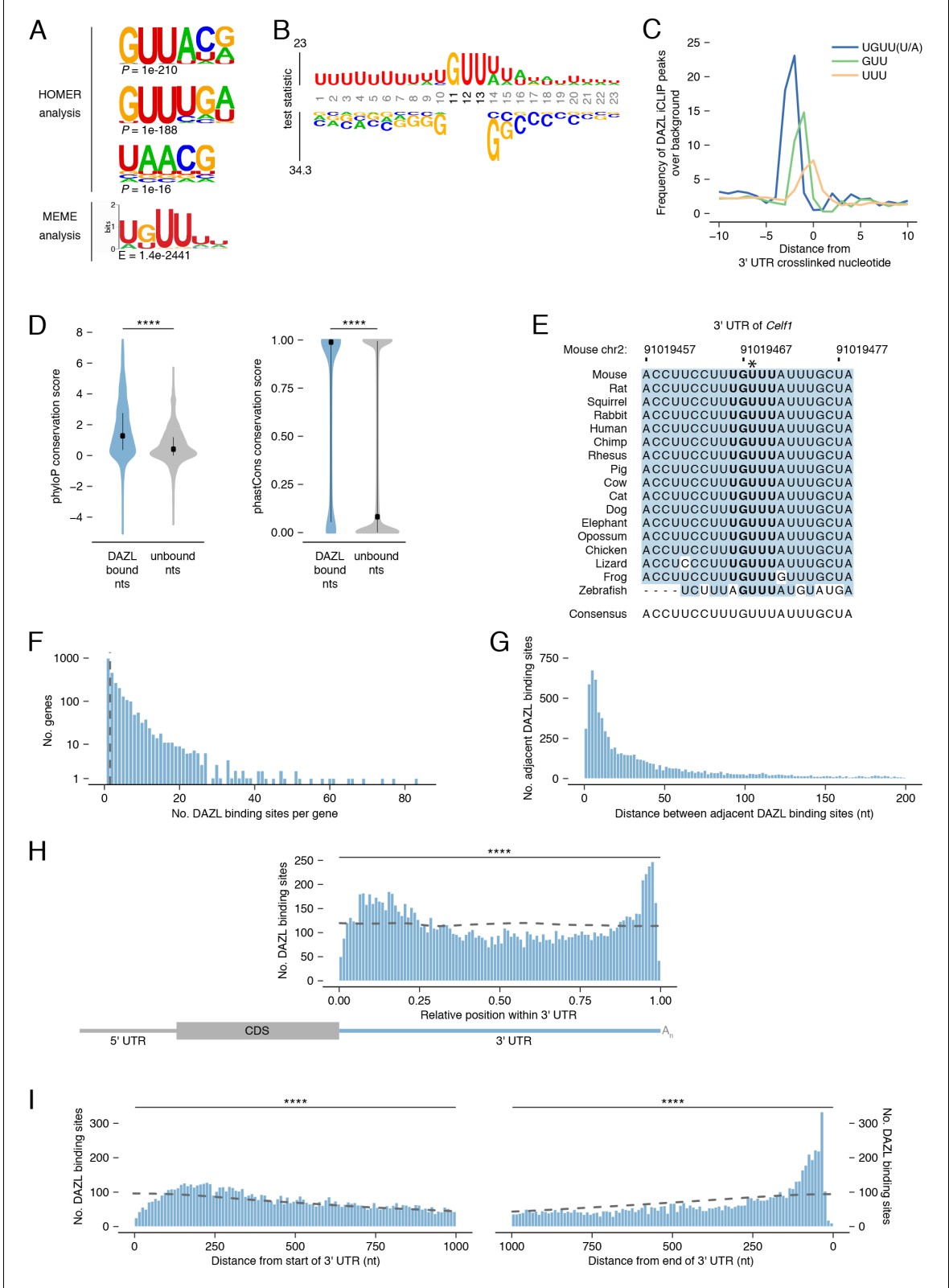

**Figure 4.** DAZL binds a UGUU(U/A) motif within 3' UTRs. (**A**) De novo motif discovery from replicated DAZL peaks in 3' UTR exons. Motif analyses were carried out with HOMER and MEME tools using crosslinked peaks ± 10 nucleotides, with all expressed 3' UTRs (TPM ≥1) as background. The top three ranked motifs identified via HOMER are shown. One statistically significant motif was identified via MEME. (**B**) GUU-centered motif analysis of replicated peaks in 3' UTRs via kpLogo. For each crosslinked peak ±10 nucleotides, the closest GUU was identified, and all sequences were aligned

*Figure 4 continued on next page*

*Figure 4 continued*

along the GUU. Background was a subset of unbound GUUs randomly selected sequences from the full-length 3′ UTRs that contain DAZL peaks. As *P* values are extremely small (<1×10$^{-308}$), residues are scaled by test statistics. (**C**) Position of all UGUU(U/A), GUU, and UUU motifs relative to crosslinked nucleotides from replicated peaks in 3′ UTRs. 0 represents the crosslinked nucleotide. Enrichment was identified relative to randomly selected sequences from the full-length 3′ UTRs that contain DAZL peaks. (**D**) Conservation of DAZL binding sites across vertebrates based on phyloP and phastCons scores. DAZL-bound nucleotides identified via iCLIP were compared with unbound nucleotides from the same 3′ UTRs (two-sided Mann-Whitney U test). (**E**) DAZL's 3′ UTR binding site in *Celf1* is conserved among vertebrates. Blue shading highlights nucleotides that reflect the consensus. Bold designates DAZL's UGUU(U/A) motif. Asterisk marks crosslinked nucleotide in DAZL iCLIP data. Sequence shown is absent from coelacanth. (**F**) Frequency of DAZL binding sites per DAZL-bound transcript. The majority of DAZL targets have more than one DAZL binding site (those targets to the right of the vertical dashed line). (**G**) Distance between adjacent DAZL binding sites in 1,649 DAZL-bound transcripts with more than one DAZL binding site. (**H**) Relative position of DAZL binding sites along the 3′ UTR. The start and end of the 3′ UTR were designated as 0 and 1, respectively. DAZL binding sites are enriched at the end and, to a lesser extent, at the start, relative to randomly selected sites in the same 3′ UTRs (dashed line) (two-sided Kolmogorov-Smirnov test). (**I**) Absolute position of DAZL binding sites along the 3′ UTR. DAZL binding sites exhibit a sharp accumulation 20–100 nucleotides from the end of the 3′ UTR and a broader accumulation 100–240 nucleotides from the start of the 3′ UTR relative to randomly selected positions within the same 3′ UTRs (dashed line) (two-sided Kolmogorov-Smirnov tests). ****, p<0.0001.

The online version of this article includes the following source data and figure supplement(s) for figure 4:

**Source data 1.** Characterization of DAZL binding within 3′ UTR.

**Figure supplement 1.** DAZL iCLIP motif enrichment and conservation.

**Figure supplement 2.** DAZL binding along the 3′ UTR.

*Sall4*, and *Sox3* (*Figure 2E*) as well as splicing factor *Celf1* (*Figure 3B*). Indeed, based on Gene Set Enrichment Analysis (GSEA), undifferentiated spermatogonial factors were overrepresented among DAZL targets with many binding sites (p<0.001; *Figure 4—figure supplement 2A*). Enriched GO categories associated with transcription and RNA splicing also showed some overrepresentation among DAZL targets, but they did not meet thresholds for statistical significance. The number of binding sites weakly correlated with transcript abundance (*Figure 4—figure supplement 2B*), which suggested that these data underestimate the actual number of binding sites for lowly expressed transcripts. The number of binding sites also weakly correlated with 3′ UTR length (*Figure 4—figure supplement 2C*), indicating that the frequency of DAZL binding on a gene is not solely driven by its 3′ UTR length. There was a similarly weak correlation between the number of binding sites and the number of UGUU(U/A) motifs in the 3′ UTR (*Figure 4—figure supplement 2D*). In part, this limited correlation is likely due to DAZL having flanking sequence preferences beyond the 5-nt motif (*Figure 4B*). For DAZL targets with multiple binding sites, we examined the distance between adjacent binding sites, and found that binding sites are positioned close together, with the median distance between peaks being 28 nt (*Figure 4G*).

We then examined the distribution of DAZL binding sites along the 3′ UTR. While DAZL binding sites were located throughout the 3′ UTR, they were strongly enriched at the start and end of the 3′ UTR (two-sided Kolmogorov-Smirnov test, p<0.0001; *Figure 4H–I*, *Figure 4—figure supplement 2E–F*). DAZL's preference for the end, but not the start, of the 3′ UTR was previously observed (*Zagore et al., 2018*). Given that the prior analysis used DAZL CLIP data from whole testes, the accumulation of binding sites at the start of the 3′ UTR in our data may be a unique feature of DAZL binding in undifferentiated spermatogonia. The accumulation of binding sites at the end of the 3′ UTR, near the poly(A) tail, is consistent with DAZL interacting with poly(A)-binding protein to regulate its targets (*Collier et al., 2005*).

In summary, DAZL interacts with the start and end of the 3′ UTR at sites that are conserved among vertebrates. Therefore, DAZL's targets in mouse are likely regulated by DAZ family proteins during spermatogenesis across vertebrates, including human.

## DAZL enhances translation of its targets

DAZL functions as a translational enhancer within the oocyte (*Collier et al., 2005*; *Sousa Martins et al., 2016*), but its molecular function during spermatogenesis remains poorly defined. We therefore asked whether DAZL similarly enhances the translation of its targets in undifferentiated spermatogonia. To isolate actively translating ribosomes specifically from germ cells, we used the RiboTag allele, which expresses HA-tagged RPL22 after Cre-mediated excision and allows for isolation of polysomes via anti-HA immunoprecipitation (IP) (*Figure 5A–B*; *Sanz et al., 2009*). To

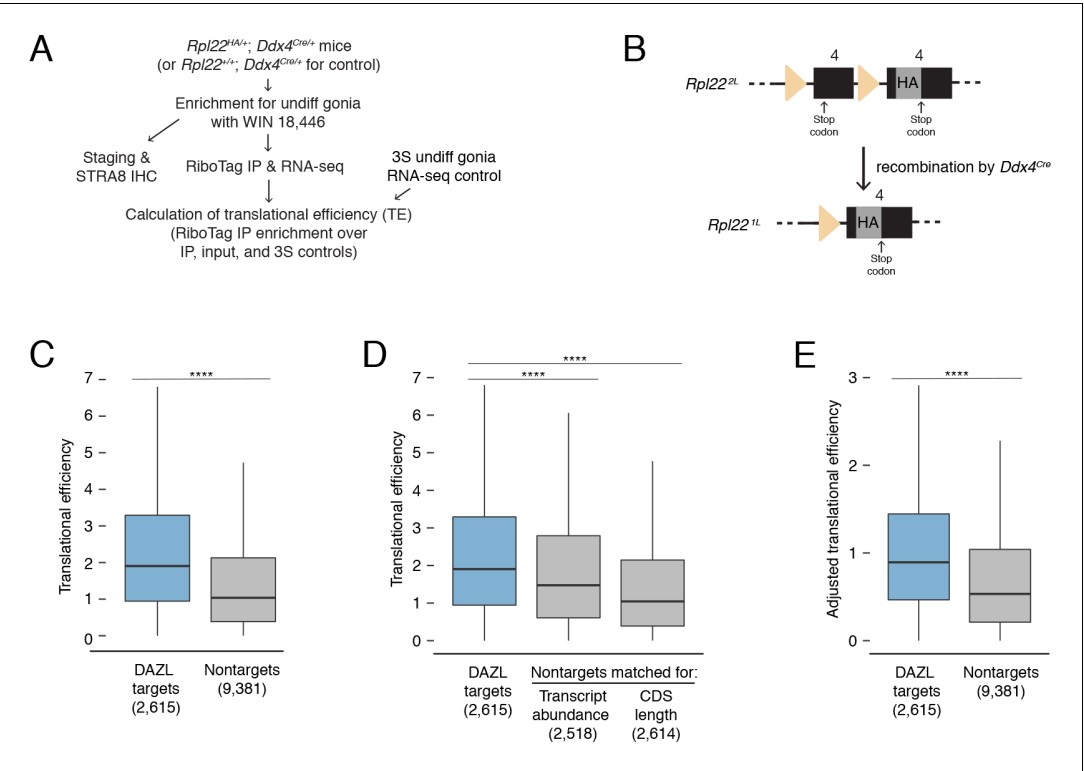

**Figure 5.** DAZL enhances translation of its targets in undifferentiated spermatogonia. (**A**) Schematic of synchronization of spermatogenesis by WIN 18,446 to enrich for undifferentiated spermatogonia for RiboTag IP and sequencing experiments. (**B**) Schematic of the RiboTag allele. The *Rpl22* locus carries a floxed exon 4, which is expressed in the absence of recombination, followed by an engineered exon four that encodes an HA tag before the stop codon. Recombination via the *Ddx4-Cre* allele removes the floxed exon four and allows for expression of the exon four that encodes the HA tag specifically in germ cells. The germ cells' ribosomes can then be immunoprecipitated via the HA tag. (**C**) Translational efficiency of DAZL targets compared with all nontargets from undifferentiated spermatogonia (two-sided Mann-Whitney U test). (**D**) Translational efficiency of DAZL targets compared with nontarget subset datasets that were sampled to match transcript abundance (TPM) and CDS length in undifferentiated spermatogonia (two-sided Mann-Whitney U test). (**E**) Adjusted translational efficiency in undifferentiated spermatogonia (two-sided Mann-Whitney U test), calculated from a multiple log-linear regression model that included transcript abundance (TPM), CDS length, 3' UTR length, codon usage (Codon Adaptiveness Index, CAI), and DAZL binding as variables. Adjusted translational efficiency was calculated by subtracting the contributions (as calculated by the model) of transcript abundance, CDS length, 3' UTR length, and codon usage from biochemically measured translational efficiency values. ****, p<0.0001.

The online version of this article includes the following source data and figure supplement(s) for figure 5:

**Source data 1.** Translational efficiencies for genes expressed in undifferentiated spermatogonia.
**Figure supplement 1.** Translational efficiency in undifferentiated spermatogonia.

recombine the RiboTag allele specifically in germ cells, we used the *Ddx4^Cre* allele. We performed RiboTag IP-seq in *Rpl22^HA/+*; *Ddx4^Cre/+* testes chemically synchronized via the 2S method for undifferentiated spermatogonia, along with controls for nonspecific binding to the antibody (RiboTag IP-seq in *Rpl22^+/+*; *Ddx4^Cre/+* 2S testes) and input RNA-seq from all IP-seq samples to control for differences in mRNA abundances among samples. We also used the RNA-seq data from 3S undifferentiated spermatogonia to control for mRNA levels in germ cells.

We measured translational efficiencies (i.e., number of ribosomes bound per transcript molecule, calculated as the enrichment in RiboTag IP-seq relative to control samples *Baser et al., 2019*) for 11,996 protein-coding genes expressed in wild-type undifferentiated spermatogonia. To verify our translational efficiencies, we examined replication-dependent histones, whose translation is coupled to S phase of the cell cycle (*Marzluff et al., 2008*). We would expect these histones to be poorly translated because our population of undifferentiated spermatogonia is not enriched for S phase.

Indeed, we confirmed that replication-dependent histones were less efficiently translated than the average transcript (two-sided Mann-Whitney U test, p=5.62×10⁻¹⁰; *Figure 5—figure supplement 1A*). With respect to translational efficiency, *Kit* fell within the bottom 10% of all genes, consistent with its translational repression in progenitor spermatogonia in the absence of retinoic acid (*Busada et al., 2015*).

We observed that DAZL targets were more efficiently translated than nontargets in undifferentiated spermatogonia (two-sided Mann-Whitney U test, p<2.2×10⁻¹⁶; *Figure 5C*). Next, we examined variables known to correlate with translational efficiency. Compared with nontargets, DAZL targets exhibited longer 3′ UTRs and less optimized codon usage (Codon Adaptation Index, CAI) (two-sided Kolmogorov-Smirnov test, p<2.2×10⁻¹⁶ for each variable; *Figure 5—figure supplement 1B*). However, these variables were associated with lower translational efficiencies (*Figure 5—figure supplement 1C*), and therefore did not contribute to the higher translational efficiencies observed in DAZL targets. DAZL targets also exhibited higher transcript abundances (TPMs) and longer coding regions than nontargets (two-sided Kolmogorov-Smirnov test, p<2.2×10⁻¹⁶ for each variable; *Figure 5—figure supplement 1B*). As these variables positively correlated with translational efficiency (*Figure 5—figure supplement 1C*), they were confounding variables in DAZL targets' higher translational efficiencies.

To control for differences in transcript abundance and coding region length, we sampled nontargets to obtain a subset that matched DAZL targets for each individual variable. We then compared translational efficiencies, and found that DAZL targets exhibited enhanced translational efficiencies compared with nontargets matched for abundance and coding region length (two-sided Mann-Whitney U test, p=1.22×10⁻¹⁴ and p<2.2×10⁻¹⁶, respectively; *Figure 5D*). When comparing the median DAZL target to the nontarget matched for transcript abundance, which was the variable most correlated with translational efficiency (*Figure 5—figure supplement 1C*), DAZL binding increased translational efficiency by 0.43 ribosomes per transcript (*Figure 5D*).

As a second statistical approach, we tested whether DAZL affects its targets' translation using a multiple log-linear model of translational efficiency that included DAZL binding and other covariates: transcript abundance, CDS length, 3′ UTR length, and codon usage. Inclusion of DAZL binding as a variable in the model significantly improved the fit to the translational efficiency data (likelihood-ratio test, p=2.97×10⁻⁴³). We then used this model to calculate adjusted translational efficiencies that control for transcript abundance, CDS length, 3′ UTR length, and codon usage. By comparing the adjusted translational efficiencies of DAZL targets and nontargets, we found that DAZL targets were more efficiently translated (two-sided Mann-Whitney U test, p<2.2×10⁻¹⁶; *Figure 5E*). DAZL binding increased the adjusted translational efficiency for the median transcript by 0.39 ribosomes per transcript (*Figure 5E*), similar to the effect size estimated by our first statistical approach (*Figure 5D*).

Therefore, based on in vivo profiling, DAZL amplifies translation in undifferentiated spermatogonia. This DAZL-bound program encompasses 20% of expressed genes in undifferentiated spermatogonia, particularly factors not specific to the testes that regulate broad cellular process such as transcription and splicing, and is required for the expansion and differentiation of progenitor spermatogonia.

## Discussion

Prior to the studies reported here, DAZL's direct functions in the spermatogonial progenitor population were not known. We demonstrate that *Dazl* promotes expansion and differentiation of spermatogonial progenitors, independent of its embryonic requirement for germ cell determination (*Figure 6*). Via biochemical analyses in undifferentiated spermatogonia in vivo, we further demonstrate that DAZL accomplishes these functions by enhancing translation of thousands of genes, including recognized spermatogonial factors that promote proliferation or differentiation of undifferentiated spermatogonia. More generally, DAZL interacts with broadly expressed, dosage-sensitive regulators of transcription and RNA metabolism, secondarily transforming the cell's transcriptome to facilitate proliferation and differentiation of spermatogonia.

Our biochemical characterization of DAZL targets in undifferentiated spermatogonia in vivo reveals a surprising breadth to the DAZL-regulated program, which encompasses ~30% of transcripts in this spermatogenic cell type. This breadth was not previously appreciated because prior studies identified DAZL targets in whole testes containing mixed stages of spermatogenesis

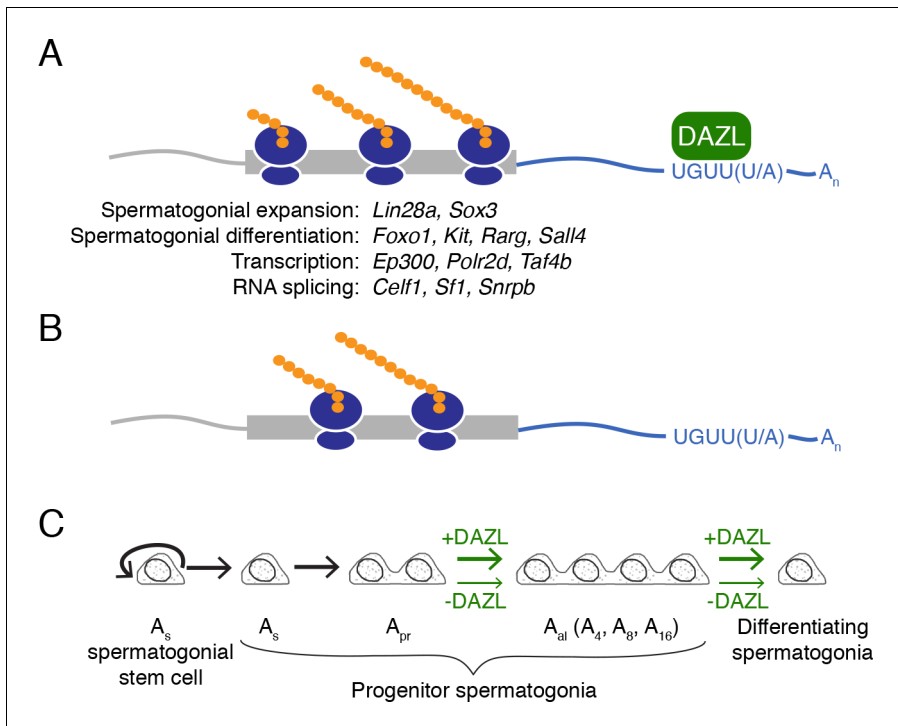

**Figure 6.** Model of DAZL's regulation of a broad translation program in undifferentiated spermatogonia. (**A**) DAZL promotes robust translation of a broad set of transcripts, ranging from those required for spermatogonial proliferation and differentiation to those that regulate fundamental cellular processes like transcription and RNA splicing. (**B**) In the absence of DAZL, transcripts normally bound by DAZL are translated less efficiently. (**C**) DAZL's broad translational program promotes the expansion of late progenitor ($A_{al}$) undifferentiated spermatogonia and subsequent spermatogonial differentiation ('+DAZL' arrows). In the absence of DAZL ('-DAZL' arrows), both expansion and differentiation occur at reduced rates.

(*Li et al., 2019*; *Zagore et al., 2018*). At the same time, this breadth suggests a role for DAZL in the regulation of global translational levels, which balances stem cell renewal with progenitor formation in other transit-amplifying populations (*Blanco et al., 2016*; *Signer et al., 2014*; reviewed in *Tahmasebi et al., 2019*; *Zismanov et al., 2016*). Specifically, hematopoietic, hair follicle, and muscle stem cells exhibit lower rates of protein synthesis compared with the progenitor cells to which they give rise (*Blanco et al., 2016*; *Signer et al., 2014*; *Zismanov et al., 2016*), and increased translation promotes the differentiation of stem cells into progenitors (*Zismanov et al., 2016*). Conversely, reduced translational levels limit progenitor formation in favor of self-renewal (*Blanco et al., 2016*). The latter phenotype is analogous to the constrained spermatogonial expansion and differentiation observed in the absence of *Dazl*. Therefore, DAZL promotes progenitor formation and function via its broad translational regulation. As DAZL promotes translation initiation, the rate-limiting step of protein synthesis during which the ribosome is assembled onto mRNA, this regulation can occur without changes in expression of the translational machinery (*Collier et al., 2005*). Increased translation of specific DAZL targets further impacts spermatogonial progenitors. DAZL-mediated expansion of the progenitor population likely requires DAZL's targeting of, for example, *Lin28a* (*Figures 2E* and *6A*), which enhances proliferation and formation of late progenitor ($A_{al}$) spermatogonia, at least in part, by blocking the biogenesis of the anti-proliferative miRNA *let-7g* (*Chakraborty et al., 2014*; *Johnson et al., 2007*; *Newman et al., 2008*; *Viswanathan et al., 2008*). Spermatogonial differentiation, which occurs in response to retinoic acid, is likely facilitated by DAZL's targeting of, for example, *Rarg* (*Figures 2E* and *6A*), a retinoic acid receptor that transduces retinoid signaling (*Gely-Pernot et al., 2012*; *Van Pelt and De Rooij, 1990*). Prior CLIP studies using whole testes identified many spermatogonial factors as DAZL targets (*Li et al., 2019*; *Zagore et al., 2018*). However, by characterizing DAZL:RNA interactions specifically in undifferentiated spermatogonia, we provide

greater insights into DAZL's binding behavior by identifying novel mRNA targets (i.e., *Rarg* and *Ep300*) and additional DAZL binding sites at previously reported targets (i.e., *Lin28a* and *Celf1*).

Investigators recently reported a different conditional deletion of *Dazl* in which undifferentiated spermatogonia are gradually lost and then depleted altogether (*Li et al., 2019*). We did not observe such a defect in stem cell maintenance in our cKO model (*Figure 1C*), likely because DAZL protein expression persisted in at least a subset of spermatogonial stem cells. The potential fragmentation of DAZL-positive $A_{pr}$ and $A_{al}$ spermatogonia into $A_s$ stem cells, which may occur under conditions of stress (*Hara et al., 2014*; *Nakagawa et al., 2007*; *Nakagawa et al., 2010*; *Zhang et al., 2016*), may have also contributed to the maintenance of a stem cell pool in our model. Using different conditional *Dazl* and *Cre* alleles, Li et al. more efficiently deleted *Dazl* in spermatogonial stem cells and thereby revealed a role for DAZL in this cell type. This study complements our own and highlights that DAZL functions at multiple points along spermatogonial development.

Because we characterized translation and DAZL's targets in undifferentiated spermatogonia from a synchronized first round of spermatogenesis, some of our results may represent the unique regulation that occurs within the first round, which originates from prospermatogonia, as opposed to later cycles of spermatogenesis, which arise from spermatogonial stem cells derived from prospermatogonia (*Hermann et al., 2015*; *Law et al., 2019*; *Yoshida et al., 2006*). Our data may also reflect the larger spermatogonial stem cell pool that is established in the neonatal testis during synchronization (*Agrimson et al., 2017*). Despite these caveats, much of the regulation identified here is likely relevant to spermatogonial function in the adult, particularly as many spermatogonial factors targeted by DAZL, such as *Lin28a* and *Rarg*, contribute to adult spermatogenesis (*Chakraborty et al., 2014*; *Gely-Pernot et al., 2012*). In addition, we genetically demonstrated DAZL's requirement in progenitor spermatogonia in adult animals, thereby supporting a role for DAZL beyond the first round of spermatogenesis.

Our in vivo findings are, in part, inconsistent with the in vitro conclusions from *Zagore et al., 2018*, who proposed that DAZL regulates transcript stability, but not translation, in spermatogonia based on work in mouse GC-1 cells (derived from immortalized Type B differentiating spermatogonia). This cell line is likely not an appropriate model for in vivo spermatogonia, as RNA-seq data (*Zagore et al., 2018*) shows low abundance of germ cell-specific markers such as *Dazl* and *Ddx4*, suggesting that these cells do not faithfully reflect the biology of germ cells. Our analyses of spermatogonia in vivo demonstrate that, for the vast majority of targets, DAZL increases translational efficiency, though we were unable to assess how the loss of DAZL translationally impacts specific genes because our conditional model's high rate of mosaicism (*Figure 1—figure supplement 2B*) obscures the effect of DAZL binding on translation. While we could not formally rule out the possibility that DAZL also regulates mRNA stability, our translational profiling of undifferentiated spermatogonia in vivo more accurately captures DAZL's molecular activity in spermatogonia.

Our refinement of DAZL's motif to UGUU(U/A) (*Figure 4A–C*) expands upon previous characterizations of DAZL's binding preferences, including a UGUU motif identified via CLIP (*Chen et al., 2014*; *Jenkins et al., 2011*; *Li et al., 2019*). However, our motif contrasts with the GUUG motif identified by another CLIP study (*Zagore et al., 2018*). This latter motif originated from CLIP-based computational analyses that fail to fully capture a protein's binding preferences (*Sugimoto et al., 2012*). By contrast, our computational analysis, which relies on iCLIP's ability to capture DAZL:RNA crosslinked sites with single-nucleotide resolution, provides a more comprehensive characterization of binding preferences (*Sugimoto et al., 2012*). In DAZL iCLIP data from this study and others (*Zagore et al., 2018*), UGUU(U/A) was more enriched at crosslinked sites than GUUG (*Figure 4— figure supplement 1A-C*) and thus is more representative of DAZL's in vivo binding preferences.

Our mechanistic characterization of DAZL function in mouse spermatogonia likely captures the activity of human DAZ, whose deletion is implicated as among the most common known genetic causes of spermatogenic failure. Like mouse DAZL, human DAZ is expressed in spermatogonia (*Menke et al., 1997*; *Nickkholgh et al., 2015*; *Szczerba et al., 2006*). At the molecular level, the two proteins' highly conserved RNA-binding domains recognize similar motifs - U(G/U)UUU by human DAZ compared with UGUU(U/A) by mouse DAZL (*Figure 4A–B*; *Dominguez et al., 2018*; *Jenkins et al., 2011*; *Saxena et al., 1996*). As mouse DAZL's 3' UTR binding sites show strict conservation among vertebrates (*Figure 4D–E*), human DAZ likely elicits similarly broad translational regulation by binding human orthologs of DAZL's murine targets. At the phenotypic level, the majority of men with reduced DAZ dosage resulting from an *AZFc* deletion form spermatozoa at a decreased

rate (*Girardi et al., 1997*; *Hopps et al., 2003*; *Kim et al., 2012*; *Nakahori et al., 1996*; *Reijo et al., 1995*; *Simoni et al., 1997*; *Vogt et al., 1996*; *Zhang et al., 2013*). This reduced spermatogenesis is consistent with the decreased spermatogonial expansion and differentiation observed in the *Dazl* cKO (*Figure 6*) and consequently points to a spermatogonial function for human DAZ. While a previous study reported that *AZFc* deletions did not affect maintenance and differentiation of human spermatogonia in vitro, culture conditions probably obviated these cells' requirement for DAZ, as wild-type spermatogonia stopped expressing DAZ protein after long term culture (*Nickkholgh et al., 2015*). Therefore, human DAZ likely promotes spermatogonial progenitor expansion and differentiation in vivo, and reduction of DAZ dosage may disrupt these developmental transitions, resulting in reduced spermatogenesis and male sterility.

In conclusion, DAZL promotes expansion and differentiation of progenitor spermatogonia by enhancing the translation of thousands of genes, including spermatogonial factors and dosage-sensitive regulators of transcription and RNA metabolism. This broad translational program in progenitor spermatogonia reflects the function of human DAZ, whose deletion is associated with one of the most common known genetic causes of spermatogenic failure.

## Materials and methods

### Animals

Mice carrying the fluorescent reporter B6D2-*Dazl*[em1(tdTomato)Huyc] (*Dazl:tdTomato*; *Nicholls et al., 2019b*) were back-crossed for four generations to a C57BL/6N (B6N) background. Mice carrying the Cre recombinase allele *Ddx4*[tm1.1(cre/mOrange)Dcp] (*Ddx4*[Cre]; *Hu et al., 2013*) were backcrossed to B6N for at least nine generations. The conditional *Dazl* allele B6N-*Dazl*[em1Dcp] (*Dazl*[2L]; *Nicholls et al., 2019b*) was generated via a CRISPR/Cas9-mediated strategy on the B6N background. Experimental animals were backcrossed to B6N for an additional two to three generations. Two *Dazl*-null alleles were used in this study: 129P2-*Dazl*[tm1Hjc] (*Dazl*[-]; *Ruggiu et al., 1997*), backcrossed to B6N for over 30 generations; and B6N-*Dazl*[em1.1Dcp] (*Dazl*[1L]; *Nicholls et al., 2019b*) which was generated by crossing animals carrying the *Dazl*[2L] allele with *Ddx4*[Cre/+] animals. These two null alleles contain deletions of largely the same exons (*Nicholls et al., 2019b*; *Ruggiu et al., 1997*). In addition, when homozygosed, both alleles produce germ cell loss on B6N animals and cause teratoma formation in 129S4/SvJae animals (*Nicholls et al., 2019b*; *Ruggiu et al., 1997*). Mice carrying the fluorescent reporter Tg(Pou5f1-EGFP)2Mnn (*Pou5f1:EGFP*; *Szabó et al., 2002*), which is a multi-copy transgene array near chromosome 9's telomere (*Nicholls et al., 2019a*), were maintained on a B6N background. Mice carrying the fluorescent Cre reporter Gt(ROSA)26Sor[tm9(CAG-tdTomato)Hze] (*ROSA26*[tdTomato]; *Madisen et al., 2010*) were backcrossed to B6N for at least 10 generations. Mice carrying the RiboTag allele B6N.129-*Rpl22*[tm1.1Psam]/J (*Rpl22*[HA] or RiboTag allele; *Sanz et al., 2009*) were maintained as homozygotes (*Rpl22*[HA/HA]). For wild-type mice, B6N mice were used. All wild-type B6N mice for experiments and backcrossing were obtained from Taconic Biosciences.

### Histology and immunostaining

Dissected tissues were fixed in Bouin's solution at room temperature for 3 hr or overnight, or in 4% (w/v) paraformaldehyde at 4 °C overnight. Fixed tissues were embedded in paraffin and sectioned to 6 µm. Slides were dewaxed in xylenes and rehydrated with an ethanol gradient. For antibody staining, antigen retrieval was carried out by boiling slides in citrate buffer (10 mM sodium citrate, 0.05% Tween 20, pH 6.0) for 10 min.

For fluorescent detection, tissue sections were treated with blocking solution (5% donkey serum in PBS for postnatal testis sections; 10% donkey serum and 2% BSA in PBS for adult testis sections), and incubated with primary antibodies diluted in blocking solution. The following primary antibodies were used: anti-DAZL (BioRad MCA2336; 1:100 dilution); anti-FOXC2 (R&D Systems AF6989; 1:250 dilution); anti-KIT (R&D Systems AF1356; 1:250 dilution); anti-mCherry (SICGEN AB0040-200; 1:300 dilution); anti-SOX9 (EMD Millipore AB5535; 1:300 dilution for postnatal testis sections and 1:250 dilution for adult testis sections); and anti-ZBTB16 (R&D Systems AF2944; 1:250 dilution). Slides were then washed in PBS and incubated with fluorophore-conjugated secondary antibodies (Jackson ImmunoResearch Laboratories). For postnatal testis tissue, slides were coverslipped with ProLong Gold Antifade reagent with DAPI (Thermo Fisher Scientific), and imaged via confocal microscopy

(Zeiss LSM 700). For adult testis tissue, slides were counterstained with DAPI, coverslipped with VEC-TASHIELD Antifade Mounting Medium for Fluorescence (Vector Laboratories), and imaged via confocal microscopy (Zeiss LSM 710).

For chromogenic immunodetection, tissue sections were stained with the ImmPRESS HRP anti-Rabbit Detection Kit (Vector Laboratories MP-7401–50) and ImmPACT DAB Peroxidase Substrate (Vector Laboratories SK-4105) using the following antibodies: anti-DAZL (Abcam ab34139; 1:200 dilution) and anti-STRA8 (Abcam ab49405; 1:500 dilution). Sections were subsequently washed with PBS and counterstained with hematoxylin.

## Image analysis

For quantification, images were obtained using a Zeiss LSM 710 NLO Laser Scanning Confocal with an LD C-Plan Apochromat 40x water objective (n.A. = 1.1). Images of each tissue section were stitched together using the Tiles tool in Zeiss ZEN Microscope Software. FOXC2, KIT, SOX9, and ZBTB16 were quantified via CellProfiler v3.1.8 (*Kamentsky et al., 2011*) using custom scripts. Tubules were scored by hand for whether or not they contained any DAZL-positive cells. The differences in number of FOXC2-positive, ZBTB16-positive, and KIT-positive cells per 100 SOX9-positive cells between conditions within the animals analyzed were assessed via two-sided Mann-Whitney U tests. The difference in KIT-positive cells per 100 ZBTB16-positive cells between conditions within the animals was assessed using a two-sided odds ratio [(KIT-positive cells/100 SOX9-positive cells) / (ZBTB16-positive cells/100 SOX9-positive cells)] (*Daniel and Cross, 2013*).

## *Isolation of* Pou5f1:*EGFP-positive spermatogonia*

Testes were dissected from control and *Dazl* cKO animals carrying the fluorescent reporter *Pou5f1*:EGFP and washed with phosphate-buffered saline (PBS). Specific genotypes can be found in *Figure 1—source data 1*. Single cell suspensions were made by removing the tunica albuginea and digesting the tissue with trypsin and 20 µg/ml DNase (Sigma, St Louis MO) in PBS. An equal volume of cold 20% fetal bovine serum (FBS) in PBS was used to terminate digestion. Cells were washed and centrifuged at 500 g and 4 °C and resuspended in 1% v/v FBS in cold PBS with DNase. The suspension was passed through a 35 µm nylon mesh filter (Corning). Single cells were gated based on forward and side light scatter. Spermatogonia were sorted based on *Pou5f1*:EGFP fluorescence using a FACSAria II (BD Biosciences) and collected in PBS.

## *RNA-seq analysis of* Dazl *cKO*

Total RNA was isolated from freshly sorted cells using TRIzol LS Reagent (Thermo Fisher Scientific) and chloroform following the manufacturer's protocol, precipitated via isopropanol with GlycoBlue Coprecipitant (Thermo Fisher Scientific AM9515), and resuspended in RNase-free water. RNA-seq libraries were prepared with the SMART-Seq v4 Ultra Low Input RNA Kit (Takara Bio) with poly(A) selection. The barcoded libraries were pooled and sequenced with 40 bp single-end reads on an Illumina HiSeq 2500 machine.

Expression levels of all transcripts in the UCSC RefSeq (refGene) transcript annotations and Retrogenes V6 annotations from the GRC38 (mm10) assembly were estimated using kallisto v0.44.0 (*Bray et al., 2016*) with sequence-bias correction using a 31 k-mer index. Quantified transcripts were filtered for mRNA, transcript-level estimated counts and transcripts per million (TPM) values were summed to the gene level, and TPMs were renormalized to transcript-per-million units. Read counts from kallisto were rounded to the nearest integer and then supplied to DESeq2 v1.26.0 (*Love et al., 2014*). After filtering for a minimum of 10 counts across all samples, Spearman rank correlation coefficients were calculated using normalized counts with the *cor* function in R and then used for hierarchical clustering. Initial hierarchical clustering revealed that one sample (control, Cre-positive, FACS batch 3) fell on its own branch, separate from the branch that contained the other six samples, so this one outlier sample was excluded from subsequent analysis.

Differential expression was analyzed in the remaining samples (n = 3 *Dazl* cKOs and n = 3 controls) in DESeq2 using default parameters. Genes were filtered for a minimum of 10 counts across all six samples. Then, the $log_2$(read counts) from each gene were modeled as a linear combination of the gene-specific effects of three variables: *Dazl* status (control or cKO), Cre status (Cre-positive or Cre-negative), and FACS batch (*Figure 1—source data 1*). Differential expression dependent on the

*Dazl* genotype was obtained from the *results* function in DESeq2 by supplying the argument: contrast = c('Dazl_genotype', 'control', 'cKO').

## Synchronization of spermatogenesis

Spermatogenesis was synchronized (for 2S and 3S samples) using a protocol originally developed by *Hogarth et al., 2013* and modified by *Romer et al., 2018*. Briefly, male mice were injected daily subcutaneously from postnatal day (P) two to P8 with WIN 18,446 (Santa Cruz Biotechnology) at 0.1 mg/gram body weight. To obtain testes enriched for undifferentiated spermatogonia, mice were euthanized on P9. For each pup, a small testis biopsy was collected for histology to confirm proper enrichment of the desired cell type, and the rest of the testes were used for iCLIP, RiboTag IP-seq, or cell sorting followed by RNA-seq. For histological verification of staging, the testis biopsy was fixed in Bouin's solution and stained for STRA8, as described above. The absence of STRA8-positive early differentiating type A spermatogonia confirmed enrichment for undifferentiated spermatogonia. All immunostaining experiments included adult testes or postnatal testes synchronized for preleptotene spermatocytes (*Romer et al., 2018*), fixed under similar conditions, as a positive control for STRA8 staining.

## Isolation of sorted (3S) undifferentiated spermatogonia

Purified undifferentiated spermatogonia for RNA-seq were obtained following the 3S protocol (*Romer et al., 2018*). Germ line lineage sorting was carried out with the *Ddx4^Cre^* and *ROSA26^tdTomato^* alleles, which contains a loxP-STOP-loxP-tdTomato construct. Via this genetic strategy, Cre recombinase was specifically expressed in germ cells, where it excised the STOP codon and activated tdTomato protein expression. After synchronization of spermatogenesis with WIN 18,446, testes were collected on P9 and biopsied for histological analysis. For the remaining testis pair, the tunica albuginea was removed, and the tissue was dissociated into a single-cell suspension using collagenase, type I (Worthington Biochemical LS004196) and trypsin as previously described (*Romer et al., 2018*) with one modification: TURBO DNAse (Thermo Fisher Scientific AM2238) was used in place of DNAse I. DAPI was added to cells prior to cell sorting on a FACSAria II (BD Biosciences). Single cells were gated based on forward and side light scatter, and high DAPI-positive (dead) cells were excluded. The undifferentiated spermatogonia were isolated based on tdTomato fluorescence and sorted in 1% BSA in PBS. Cells were transferred into and stored at −80 °C in Trizol LS (Thermo Fisher Scientific).

## RNA-seq analysis of sorted (3S) undifferentiated spermatogonia

For RNA extraction, the undifferentiated spermatogonia from a single chemically synchronized postnatal animal were used as one biological replicate; two biological replicates were carried out in total. RNA was extracted using Trizol LS (Thermo Fisher Scientific) and chloroform following the manufacturer's protocol, precipitated via isopropanol with GlycoBlue Coprecipitant (Thermo Fisher Scientific AM9515), and resuspended in RNase-free water. RNA-seq libraries were prepared with the TruSeq Stranded mRNA kit (Illumina) with RiboZero depletion. The barcoded libraries were pooled and sequenced with 40 bp single-end reads on an Illumina HiSeq 2500 machine.

Expression levels of all transcripts in the UCSC RefSeq (refGene) transcript annotations and Retrogenes V6 annotations from the GRC38 (mm10) assembly were estimated using kallisto v0.44.0 (*Bray et al., 2016*) with sequence-bias correction using a 31 k-mer index.

For the comparison of biological replicates and iCLIP analysis, rRNA transcripts were removed, and transcript-level estimated counts and TPM values were summed to the gene level, with protein-coding transcripts and non-coding transcripts from the same gene summed separately, and TPM values were renormalized to transcript-per-million units.

For the comparison of sorted undifferentiated spermatogonia to other spermatogonial datasets, quantified transcripts were filtered for mRNAs (both protein-coding transcripts and non-coding transcripts), and rRNA transcripts were excluded. Transcript-level estimated counts and TPM values were summed to the gene level, with protein-coding and non-coding transcripts from the same gene summed together, and TPM values were renormalized to transcript-per-million units. The following RNA-seq datasets were also processed using these methods: NCBI GEO GSE102783 (samples GSM2746356 and GSM2746357 only) *Maezawa et al., 2017*; DDBJ/GenBank/EMBL

DRA002477 (samples DRR022939 and DRR022945 only) *Kubo et al., 2015*; and NCBI GEO GSE107124 (*La et al., 2018b*). Read counts from kallisto were rounded to the nearest integer and the default procedure in DESeq2 v1.26.0 (*Love et al., 2014*) was applied to normalize read counts across samples. Spearman rank correlation coefficients were calculated using normalized counts with the *cor* function in R and then used for hierarchical clustering.

## iCLIP library construction, sequencing, and computational analysis

After synchronization of spermatogenesis in wild-type animals via the 2S protocol, testis tubules were dissociated by pipetting in ice-cold PBS and irradiated three times at 200 mJ/cm$^2$ at 254 nm in a Stratalinker 2400. Irradiated testis tubules were then pelleted, supernatant was removed, and samples were stored at −80 ˚C. iCLIP libraries were prepared as previously described (*Huppertz et al., 2014*), with the following modifications: the thawed testis tissue from one animal was prepared in 640 μl lysis buffer, lysates were digested with 1.0 U Turbo DNase (Thermo Fisher Scientific AM2238) and 0.01 U RNase I (Thermo Fisher Scientific AM2295) per 500 μL lysate for 3 min at 37 ˚C and 1100 rpm on a Thermomixer R (Eppendorf), antibody [anti-DAZL (Abcam ab34139) or IgG isotype control (Santa Cruz Biotechnologies sc-2027)] was conjugated to Dynabeads Protein G (Thermo Fisher Scientific 10003D) at a ratio of 10 μg antibody per 100 μl beads, and immunoprecipitation was carried out using 100 μl antibody-conjugated beads per 500 μl lysate. During preliminary experiments on unsynchronized P20 testes to optimize RNase digestion, lysates were treated with 1.0, 0.1, or 0.01 U RNase I (Thermo Fisher Scientific AM2295) and 1.0 U Turbo DNase (Thermo Fisher Scientific AM2238) per 500 μL lysate for 3 min at 37 ˚C and 1100 rpm on a Thermomixer R (Eppendorf), and radiolabeled RNA fragments were obtained from RNA:protein complexes as previously described (*Huppertz et al., 2014*), run on a 6% Novex TBE-Urea gel (Thermo Fisher Scientific EC6865BOX), transferred to Amersham Hybond-XL nylon membrane, and exposed to film overnight at −80 ˚C. Prior to library construction, the efficiency of reverse transcription primers was verified. DAZL (n = 3) and IgG (n = 3) iCLIP libraries were pooled and sequenced with 40 bp single-end reads on an Illumina HiSeq 2500 machine.

The 5′ end of each raw iCLIP read contained (from 5′ to 3′) a three nt random barcode, followed by a four nt sample-specific barcode, followed by a two nt random barcode. Reads were quality trimmed using Cutadapt v.1.8 (options: -q 20 m 24) (*Martin, 2011*). Using the FASTX-Toolkit v.0.0.14 (http://hannonlab.cshl.edu/fastx_toolkit/index.html), PCR duplicates were collapsed (fastx_collapser tool), the 5′-most three nt random barcodes were removed from each read (fastx_trimmer tool with option -f 4), the libraries were demultiplexed via the sample-specific barcodes (fastx_barcode_splitter.pl tool with option –bol), and the sample-specific and remaining random barcodes were removed (fastx_trimmer tool with option -f 7). iCLIP libraries and RNA-seq libraries from sorted (3S) undifferentiated spermatogonia were mapped to the mouse genome (mm10) via STAR v.2.5.4b (*Dobin et al., 2013*) (options: `−−outFilterMultimapNmax` 1 `−−alignEndsType` Extend5pOfRead1 `−−outFilterMismatchNmax` 2 `−−outSAMattributes` None). All other parameters were set to default. The iCLIP mapped reads were then converted to crosslinked nucleotides, defined as the nucleotide immediately preceding the first nucleotide of the mapped read as identified by truncated iCLIP cDNAs (*König et al., 2010*). DAZL crosslinked peaks were then called from DAZL iCLIP crosslinked nucleotides via ASPeak v.2.0.0 (*Kucukural et al., 2013*) using the IgG iCLIP crosslinked nucleotides for the -control parameter and the RNA-seq mapped reads for the -rnaseq parameter. All other parameters were set to default. The UCSC RefSeq (refGene) transcript annotations and Retrogenes V6 annotations from the GRC38 (mm10) assembly were used to call peaks. After preliminarily assessing the genomic distribution of DAZL crosslinked peaks, we used the following hierarchy: 3′ UTR exon > 5′ UTR exon > coding exon > ncRNA > retrogene > intron > intergenic region. Called peaks were filtered for FDR < 0.05. Replicated peaks were defined as crosslinked nucleotides that were present in at least two of three biological replicates. To identify the replicated peaks that were most biologically relevant to undifferentiated spermatogonia, replicated 3′ UTR peaks that were filtered for genes expressed at a minimum of 1 TPM in RNA-seq data from sorted (3S) undifferentiated spermatogonia (*Figure 2—source data 2*).

## Enrichment analysis

We identified 58 factors that regulate the development and differentiation of undifferentiated spermatogonia (reviewed in *Mecklenburg and Hermann, 2016*). We limited this list to those factors with functional evidence (knockout or knockdown data from a mouse or cell culture model). To this list, we added two factors whose functional roles in undifferentiated spermatogonia were reported after the publication of this review, *Foxc2* (*Wei et al., 2018*) and *Tsc22d3* (also known as *Gilz*) (*La et al., 2018a*). The complete list of factors is presented in *Figure 2—source data 1*. To determine whether factors that regulate undifferentiated spermatogonia appear more frequently among DAZL targets than among all genes expressed in undifferentiated spermatogonia, these groups were statistically compared using a one-tailed hypergeometric test via the *phyper* function with *lower.tail = F* in R.

GO analysis of DAZL 3' UTR targets was carried out using PANTHER v.13.1 (*Mi et al., 2017*) with the GO Slim Biological Processes annotation with overrepresentation test (*Figure 3—source data 1*). The background gene set was those genes that are expressed in sorted (3S) undifferentiated spermatogonia (TPM $\geq$1). Statistical significance was calculated via Fisher's exact test with Bonferroni correction.

## Analysis of testis-specific factors and expression breadth

Testis-specific factors and expression breadth were characterized using RNA-seq data from 12 mouse tissues collected in a single study (NCBI GEO GSE125483; *Naqvi et al., 2019*). Expression levels of all transcripts in the UCSC RefSeq (refGene) transcript annotations and Retrogenes V6 annotations from the GRC38 (mm10) assembly were estimated using kallisto v.0.44.0 (*Bray et al., 2016*) with sequence-bias correction using a 31 k-mer index. Quantified transcripts were filtered for mRNAs, transcript-level estimated counts and TPM values were summed to the gene level, and TPM values were renormalized to transcript-per-million units.

A gene was identified as testis-specific if (i) the gene was expressed at a minimum of 5 TPM in the testis and (ii) at least 25% of the gene's $\log_2$ normalized expression summed across all 12 tissues came from the testis. A complete list of testis-specific genes is in *Figure 3—source data 2*. To determine if DAZL targets less are frequently testis-specific than nontargets, we used a one-tailed hypergeometric test via the *phyper* function with *lower.tail = T* in R.

For each gene, expression breath was calculated as the mean of the 12 tissue-specific $\log_2$ expression values normalized by the maximum expression value for that gene across the 12 tissues (*Figure 3—source data 2*). To determine if DAZL targets have a greater expression breadth than nontargets, we used a one-sided Mann-Whitney U test via the *wilcox.test* function with *alternative = 'greater'* in R.

## Mouse-yeast ortholog analysis

Mouse-yeast orthologs were downloaded from Ensembl 98. Mouse genes were filtered by gene name for those that were expressed in sorted (3S) undifferentiated spermatogonia, and any mouse gene with any type of orthology in yeast was denoted as having a yeast ortholog (*Figure 3—source data 3*). To determine if genes with yeast orthologs appear more frequently among DAZL targets than among all genes expressed in undifferentiated spermatogonia, these groups were statistically compared using a one-tailed hypergeometric test via the *phyper* function with *lower.tail = F* in R.

## Purifying selection analysis

Mouse-human orthologs and their non-synonymous and synonymous substitution rates (dN and dS, respectively) were obtained from Ensembl 98. Mouse genes were filtered for (i) 1:1 mouse:human orthologs ('orthology_one2one') and (ii) the most robustly expressed isoform (RefSeq mRNA ID) per expressed gene in sorted (3S) undifferentiated spermatogonia. Transcript IDs were removed, and non-unique lines were collapsed, resulting in each line representing a unique gene. For each pair of orthologs, the ratio of non-synonymous to synonymous substitution rates (dN/dS) was calculated in R (*Figure 3—source data 4*). 'NA' dN/dS values were removed. To determine if DAZL targets have a lower dN/dS ratio than nontargets, a one-sided Mann-Whitney U test was applied using the *wilcox.test* function with *alternative='less'* in R.

## Haploinsufficiency and human genic copy number variation analysis

The Exome Aggregation Consortium (ExAC) data was downloaded from (ftp://ftp.broadinstitute.org/pub/ExAC_release/release0.3.1/cnv/). Haploinsufficiency predictions with imputation for human genes were downloaded from Dataset S2 of *Huang et al., 2010* (https://doi.org/10.1371/journal.pgen.1001154.s002). For both datasets, human genes were filtered for those with 1:1 mouse:human orthologs, as annotated in Ensembl 98, and then converted to mouse gene names. Genes were then filtered for those expressed in sorted (3S) undifferentiated spermatogonia (*Figure 3—source data 5*). To determine if the human orthologs of DAZL targets are more intolerant of copy number variation, deletions, and duplications or exhibited greater haploinsufficiency than those of nontargets, a one-sided Mann-Whitney U test was applied using the *wilcox.test* function with parameter *alternative='greater'* in R.

## Conserved miRNA targeting analysis

Pre-calculated $P_{CT}$ scores for all gene-miRNA family interactions were obtained from TargetScan-Mouse v7.1 (http://www.targetscan.org/mmu_71/mmu_71_data_download/Summary_Counts.all_predictions.txt.zip) (*Agarwal et al., 2015*) and filtered for mouse miRNAs. Genes were filtered for those expressed in sorted (3S) undifferentiated spermatogonia. For each gene, null $P_{CT}$ scores were removed, and the mean of the remaining $P_{CT}$ scores was calculated (*Figure 3—source data 5*). To determine if DAZL targets have higher $P_{CT}$ scores than nontargets, a one-sided Mann-Whitney U test was applied using the *wilcox.test* function with parameter *alternative='greater'* in R.

## Motif analysis

De novo motif analysis of replicated peaks in 3' UTR exons was carried out using HOMER v4.9.1 (*Heinz et al., 2010*) (options: -size 21 S 10 -len 3,4,5,6 -rna) and MEME v4.11.2 (*Bailey and Elkan, 1994*) (options: -rna -mod oops -nmotifs 6 -minw 3 -maxw 6 -maxsize 2000000 with a 0-order background model). The 3' UTR sequences from genes expressed at a minimum TPM of 1 in sorted (3S) undifferentiated spermatogonia were used as background. If a gene had multiple 3' UTR isoforms, the 3' UTR from the most robustly expressed isoform (as identified via kallisto) was used.

The GUU-centered motif analysis was carried out via kpLogo v1.0 (*Wu and Bartel, 2017*) (options: -weighted). For each replicated crosslinked nucleotide ±10 nt, the closest GUU motif was identified, extended by 10 nt on each side, and assigned a weight of –log(*P* value) from the associated crosslinked nucleotide. As background, GUUs (±10 nt) that are not DAZL bound were randomly selected from the 3' UTRs of DAZL-bound transcripts, for a total of 14,381 control sequences, and assigned a weight of 0.

To study motif frequency relative to DAZL crosslinked sites, the starting position of each motif relative to each crosslinked nucleotide was identified. All DAZL iCLIP crosslinked nucleotides were used for this analysis. Motif frequency at crosslinked sites was then normalized to a background motif frequency, estimated from a set of control sequences comprised of 10–20 randomly-selected sequences per DAZL-bound 3' UTR, for a total of 51,866 control sequences. For each gene, the most robustly expressed 3' UTR isoform, as described above for de novo motif analysis, was used.

To calculate the percentage of binding sites with a particular motif, a binding site was scored as containing the motif if the 5'-most nucleotide of that motif overlapped with a binding site's 5'-most crosslinked site ±10 nt. The nucleotide frequency within DAZL-bound 3' UTRs (using the most robustly expressed isoform for each gene) was calculated using MEME Suite's fasta-get-markov tool (*Bailey and Elkan, 1994*) (options: -rna -m 0), and this nucleotide frequency was used to calculate the expected percentage of each motif within a random 21 nt sequence.

Motif enrichment at replicated peaks from each genomic region was assessed via the MEME Suite's AME v4.11.2 (*McLeay and Bailey, 2010*) (options: `-scoring` avg `-method` ranksum) using shuffled control sequences. To limit the analysis to instances where the two U's following G in each motif overlapped the crosslinked nucleotides, the length of the input sequences were adjusted based on motif length. For enrichment of GUU and UGUU(U/A) motifs, the replicated crosslinked nucleotides ± 2 nt and ±3 nt, respectively, were used. Bonferonni correction was applied for multiple testing.

For motif analysis of DAZL iCLIP data from P6 testes (NCBI GEO GSE108183; *Zagore et al., 2018*), peaks were called from raw sequencing data as described above, but without RNA-seq data

for abundance-sensitive detection of peaks. Motif enrichment at replicated 3' UTR peaks was assessed via AME as described above. For GUUG, GUUC, and UUU(C/G)UUU enrichment, the replicated crosslinked nucleotides ± 2 nt, ±2 nt, and ±5 nt, respectively for each motif, were used. Bonferonni correction was applied for multiple testing.

Sequence conservation analysis phyloP and phastCons scores (*Pollard et al., 2010*; *Siepel et al., 2005*), calculated from the multiple genome-wide alignments of 60 vertebrate species, were downloaded for the GRC38 (mm10) assembly from the 'phyloP60wayAll' and 'phastCons60way' tables under the 'Conservation' track in the UCSC Genome Browser. Replicated DAZL-crosslinked nucleotides in 3' UTRs were compared to noncrosslinked nucleotides from the same 3' UTRs using a two-sided Mann-Whitney U test, applied via the *wilcox.test* function in R. For each gene, the 3' UTR from the most robustly expressed isoform, as described above for de novo motif analysis, was used. To visualize sequence conservation at specific DAZL binding sites, the multiple alignments at specific sites were downloaded for the GRC38 (mm10) assembly from the 'multiz60way' table under the 'Conservation' track in the UCSC Genome Browser. To assess conservation across vertebrates, we examined the following species: *Mus musculus* (mouse), *Rattus norvegicus* (rat), *Spermophilus tridecemlineatus* (squirrel), *Oryctolagus cuniculus* (rabbit), *Homo sapiens* (human), *Pan troglodytes* (chimp), *Macaca mulatta* (rhesus), *Sus scrofa* (pig), *Bos taurus* (cow), *Felis catus* (cat), *Canis lupus familiaris* (dog), *Loxodonta africana* (elephant), *Monodelphis domestica* (opossum), *Gallus gallus* (chicken), *Anolis carolinensis* (lizard), *Xenopus tropicalis* (frog), *Latimeria chalumnae* (coelacanth), and *Danio rerio* (zebrafish).

## Gene Set Enrichment Analysis (GSEA)

DAZL targets were ranked by number of DAZL binding sites in descending order and analyzed via the 'GSEAPreranked' tool from GSEA v4.0.3 (*Mootha et al., 2003*; *Subramanian et al., 2005*) using a custom gene matrix containing three gene sets: one set of genes that regulate undifferentiated spermatogonia (*Figure 2—source data 1*) and two sets of genes from GO terms 'mRNA splicing, via spliceosome' (GO:0000398) and 'transcription by RNA polymerase II' (GO:0006366), downloaded from Mouse Genome Informatics v6.15 (http://www.informatics.jax.org/).

## Positional analysis along the 3' UTR

For DAZL binding sites that consisted of more than one consecutive crosslinked nucleotide, the 5' most crosslinked nucleotide on the transcript was used. For genes with multiple isoforms, the most robustly expressed isoform per gene in sorted (3S) undifferentiated spermatogonia was used. The relative and absolute positions within 3' UTRs were calculated via MetaPlotR (*Figure 4—source data 1*; *Olarerin-George and Jaffrey, 2017*). A background distribution independent of sequence context was modeled by randomly generating one 3' UTR position for each DAZL binding site on that transcript (*Figure 4—source data 1*). A background distribution of UGUU(U/A) motifs was identified in the DAZL-bound 3' UTRs using a custom script (*Figure 4—source data 1*). Distributions were compared using a two-sided Kolmogorov-Smirnov test, as applied by the *ks.test* function in R.

## Preparation of ribosome occupancy sequencing libraries

$Rpl22^{HA/+}$; $Ddx4^{Cre/+}$ animals were used for the RiboTag immunoprecipitation (n = 3), and $Rpl22^{HA/+}$; $Ddx4^{+/+}$ littermates were used as controls for nonspecific binding to the antibody (n = 3). Polysomes were immunoprecipitated using the RiboTag as previously described (*Sanz et al., 2009*), with some modifications. Postnatal testes chemically synchronized for undifferentiated spermatogonia were homogenized at 2% w/v in polysome buffer (50 mM Tris, pH 7.4, 100 mM KCl, 12 mM $MgCl_2$, 1% Nonidet P-40, 1 mM DTT, 200 U/mL Promega RNasin, 1 mg/mL heparin, 100 μg/mL cycloheximide, Sigma cOmplete EDTA-free protease inhibitor cocktail), first by gently pipetting, then by gently pulling through a long 26 gauge needle. Homogenate was centrifuged at 10,000 g for 10 min at 4 °C, and supernatant was transferred to a fresh tube. 35.0 μL of each sample was set aside as an input control. 2.5 μL anti-HA (BioLegend #901513; previously Covance #MMS-101R) was added to 400 μL of remaining lysate, and samples were gently rotated for 4 hr at 4 °C. Dynabeads Protein G (Thermo Fisher Scientific 10003D) were washed three times in homogenization buffer (50 mM Tris, pH 7.4, 100 mM KCl, 12 mM $MgCl_2$, 1% Nonidet P-40), resuspended in original volume, and aliquoted 200 μL per tube. After the final wash was removed from the Dynabeads, the lysate with

antibody was added to the Dynabeads and incubated with gentle rotation overnight at 4 °C. The following day, the Dynabeads were washed three times for 10 min each with gentle rotation at 4 °C in high salt buffer (50 mM Tris, pH 7.4, 300 mM KCl, 12 mM MgCl$_2$, 1% Nonidet P-40, 1 mM DTT, 100 μg/mL cycloheximide).

To extract the RNA, the final wash was removed from the Dynabeads and replaced with 350 μl of high salt buffer supplemented with 1% SDS and 0.25 μg/μL Proteinase K, RNA grade (Thermo Fisher Scientific 25530049). Samples were incubated at 37 °C for 30 min with gentle mixing. Samples were then mixed with an equal volume of acid phenol:chloroform:IAA, pH 4.5, and using phase lock gel tubes, the RNA-containing aqueous phase was transferred to a new tube. RNA was ethanol-precipitated using GlycoBlue Coprecipitant (Thermo Fisher Scientific AM9515), resuspended in RNase-free water, and quantified via Agilent Bioanalyzer 2100. 120–160 ng total RNA was isolated from Ribo-Tag-positive samples while 1–3 ng total RNA was isolated from RiboTag-negative controls, and 600–1000 ng total RNA was isolated from input controls. All samples were prepared as RNA-seq libraries with the SMARTer Stranded Total RNA-Seq Kit v2 – Pico Input (for 250 pg – 100 ng input) (Takara Bio) with RiboZero depletion. The barcoded libraries were pooled and sequenced with 50 bp single-end reads on an Illumina HiSeq 2500 machine.

## Analysis of ribosome occupancy sequencing data

Raw sequencing data was trimmed for quality using Cutadapt and pseudoaligned using kallisto, as previously described. Quantified transcripts were filtered for coding mRNA only, transcript-level estimated counts and transcripts per million (TPM) values were summed to the gene level, and TPMs were renormalized to transcript-per-million units. The sorted (3S) undifferentiated spermatogonia RNA-seq data were similarly reprocessed. Read counts from kallisto were rounded to the nearest integer and then supplied to DESeq2 v.1.22.2 (*Love et al., 2014*). DESeq2's default procedure was applied to normalize read counts across samples. Data were analyzed with multi-factor designs to estimate protein-specific binding over controls. The dataset was filtered to those genes with at least 15 normalized counts in RiboTag IP data (n = 3) and at least 10 normalization counts in RNA-seq data from sorted (3S) undifferentiated spermatogonia.

Translational efficiency was calculated as the RiboTag IP-specific enrichment over the RiboTag input and sorted (3S) undifferentiated spermatogonia samples as well as over the control IP samples, which reflect nonspecific antibody binding (*Baser et al., 2019*). Specifically, the log$_2$(read counts) for each gene was modeled as a linear combination of the gene-specific effects of three variables: binding to the RiboTag protein ('RiboTag.specific'), nonspecific binding to the anti-HA antibody ('RiboTag.nonspecific'), and germ cell specific-transcriptome ('germ.cell.specific') (*Figure 5—source data 1*). RiboTag IP-specific enrichment was obtained from the *results* function in DESeq2 by supplying the argument: contrast = c('Ribotag.specific', '1', '0'). We obtained translational efficiency values for a total of 11,996 genes. All of these genes were expressed at a minimum of 1 TPM (*Figure 5—source data 1*).

For the histone analysis, the gene list from GO term 0006335 'DNA replication-dependent nucleosome assembly' (*Ashburner et al., 2000*; *The Gene Ontology Consortium, 2019*; *The Gene Ontology Consortium, 2019*) was downloaded from AmiGO 2 v.2.5.12, last updated 2019-07-02 (DOI 10.5281//zenodo.3267438) and filtered for histone genes. Histone genes were compared to non-histone genes via a two-sided Mann-Whitney U test, applied via the *wilcox.test* function in R.

The translational efficiencies of DAZL targets and all nontargets were compared via a two-sided Mann-Whitney U test, applied via the *wilcox.test* function in R.

To quantify biased codon usage, the Codon Adaptiveness Index (CAI) (*Sharp and Li, 1987*) was calculated for all expressed transcripts using coRdon v1.2.0 (*Elek et al., 2019*), with relative adaptiveness (w) computed from a reference sequence set comprised of the top 5% of expressed transcripts in sorted (3S) undifferentiated spermatogonia. The coding region of the most robustly expressed isoform per expressed gene in sorted (3S) undifferentiated spermatogonia was used (*Figure 5—source data 1*).

Correlations between log$_2$-transformed translational efficiency and log$_2$-transformed transcript abundance (TPM), CDS length, 3' UTR length, and CAI were calculated using Spearman rank correlation coefficients via the *cor* function in R.

The transcripts not bound by DAZL were sampled to produce datasets that matched the DAZL targets in their transcript abundance (TPM) and CDS length (*Figure 5—source data 1*). For genes

with multiple expressed isoforms, the coding region of the most robustly expressed isoform was used for CDS length. The translational efficiencies of DAZL targets were compared to sampled non-targets via a two-sided Mann-Whitney U test, applied via the *wilcox.test* function in R.

For linear regression analysis, the correlation between DAZL binding and $log_2$(translational efficiency) was calculated via the point biserial coefficient. All other coefficients from log-transformed variables were calculated via Pearson's correlation coefficient. The multiple log-linear regression analysis was carried out using the *lm* function in R. Translational efficiency was modeled as follows:

$$log_2(translational\ efficiency) \sim intercept + log_2(TPM) + log_2(CDS\ length) + \\ log_2(3'\ UTR\ length) + log_2(CAI) + DAZL\ binding$$

For CDS length and 3' UTR length, the most robustly expressed isoform was used. To verify that the addition of each variable to the model improved the fit to the data, we performed a likelihood ratio test comparing the model fit with and without that variable in a stepwise fashion following the order of the variables listed above. The adjusted translational efficiencies of DAZL targets and all nontargets were compared via a two-sided Mann-Whitney U test, applied via the *wilcox.test* function in R.

## Data and code availability

All sequencing data generated in this study are available at NCBI Gene Expression Omnibus (accession number GSE145177). Code is available on GitHub for image analysis (https://github.com/mmikedis/Analysis-of-tubule-cross-sections-via-CellProfiler; *Mikedis, 2020a*; copy archived at https://github.com/elifesciences-publications/Analysis-of-tubule-cross-sections-via-CellProfiler), iCLIP analysis (https://github.com/mmikedis/iCLIP_analysis; *Mikedis, 2020b*; copy archived at https://github.com/elifesciences-publications/iCLIP_analysis), and estimating translational efficiency (https://github.com/mmikedis/RiboTag_translational_efficiency; *Mikedis, 2020c*; copy archived at https://github.com/elifesciences-publications/RiboTag_translational_efficiency).

## Acknowledgements

We thank Y-C Hu for *Dazl:tdTomato* reporter mice; M Goodheart for technical support; H Christensen, ML Kojima, and KA Romer for experimental assistance; T-J Cho for advice on library preparation; DW Bellott, AK Godfrey, S Naqvi, and H Skaletsky, as well as G Bell (Whitehead Bioinformatics and Researching Computing Core) for advice and assistance on the bioinformatics analysis; T Eisen for feedback on the translational efficiency analysis; and H Christensen, DW Bellott, JF Hughes, and K Xiang for comments on the manuscript. We thank the organizers of the 2015 EMBO practical course 'iCLIP: Genomic views of protein-RNA interactions' for sharing their expertise on the iCLIP protocol. We acknowledge the technical expertise of the Whitehead Institute FACS, Keck Imaging, and Genome Technology Core facilities.

## Additional information

### Funding

| Funder | Grant reference number | Author |
|---|---|---|
| Howard Hughes Medical Institute | Page laboratory | David C Page |
| Lalor Foundation | Postdoctoral fellowship | Maria M Mikedis |
| Eunice Kennedy Shriver National Institute of Child Health and Human Development | F32HD093391 | Maria M Mikedis |
| National Natural Science Foundation of China | 81471507 | Yuting Fan |
| National Key Research and Development Program of China | 2017YFC1001600 | Yuting Fan |

| Hope Funds for Cancer Research | HFCR-15-06-06 | Peter K Nicholls |
| National Health and Medical Research Council | GNT1053776 | Peter K Nicholls |

The funders had no role in study design, data collection and interpretation, or the decision to submit the work for publication.

### Author contributions
Maria M Mikedis, Conceptualization, Data curation, Formal analysis, Funding acquisition, Investigation, Visualization, Writing - original draft, Writing - review and editing; Yuting Fan, Peter K Nicholls, Conceptualization, Resources, Funding acquisition, Investigation, Methodology, Writing - review and editing; Tsutomu Endo, Dirk G de Rooij, Formal analysis, Investigation, Writing - review and editing; Emily K Jackson, Formal analysis, Writing - review and editing; Sarah A Cobb, Investigation, Writing - review and editing; David C Page, Conceptualization, Resources, Supervision, Funding acquisition, Writing - review and editing

### Author ORCIDs
Maria M Mikedis https://orcid.org/0000-0001-7800-7120
Yuting Fan https://orcid.org/0000-0002-2143-0187
Peter K Nicholls http://orcid.org/0000-0002-5540-442X
Tsutomu Endo https://orcid.org/0000-0001-9394-362X
Emily K Jackson https://orcid.org/0000-0002-2972-4139
Dirk G de Rooij http://orcid.org/0000-0003-3932-4419
David C Page http://orcid.org/0000-0001-9920-3411

### Ethics
Animal experimentation: All experiments involving mice were performed in accordance with the guidelines of the Massachusetts Institute of Technology (MIT) Division of Comparative Medicine, which is overseen by MIT's Institutional Animal Care and Use Committee (IACUC). The animal care program at MIT/Whitehead Institute is accredited by the Association for Assessment and Accreditation of Laboratory Animal Care, International (AAALAC), and meets or exceeds the standards of AAALAC as detailed in the Guide for the Care and Use of Laboratory Animals. The MIT IACUC approved this research (no. 0617-059-20).

### Decision letter and Author response
Decision letter https://doi.org/10.7554/eLife.56523.sa1
Author response https://doi.org/10.7554/eLife.56523.sa2

## Additional files
### Supplementary files
• Supplementary file 1. Information on statistical tests, alternative hypotheses, and *P* values used.
• Transparent reporting form

### Data availability
All sequencing data generated in this study are available at NCBI Gene Expression Omnibus accession number GSE145177.

The following datasets were generated:

| Author(s) | Year | Dataset title | Dataset URL | Database and Identifier |
|---|---|---|---|---|
| Fan Y, Mikedis MM, Page DC | 2020 | Gene expression in Pou5f1:EGFP-positive spermatogonia from Dazl conditional knockout and control | https://www.ncbi.nlm.nih.gov/geo/query/acc.cgi?acc=GSE144919 | NCBI Gene Expression Omnibus, GSE144919 |

males

| Mikedis MM, Page DC | 2020 | Gene expression in undifferentiated spermatogonia | https://www.ncbi.nlm.nih.gov/geo/query/acc.cgi?acc=GSE144923 | NCBI Gene Expression Omnibus, GSE144923 |
| Mikedis MM, Page DC | 2020 | DAZL targets in undifferentiated spermatogonia | https://www.ncbi.nlm.nih.gov/geo/query/acc.cgi?acc=GSE144920 | NCBI Gene Expression Omnibus, GSE144920 |
| Mikedis MM, Page DC | 2020 | Translational profiling in undifferentiated spermatogonia | https://www.ncbi.nlm.nih.gov/geo/query/acc.cgi?acc=GSE144922 | NCBI Gene Expression Omnibus, GSE144922 |
| Mikedis MM, Page DC | 2020 | DAZL mediates a broad translational program regulating expansion and differentiation of spermatogonial progenitors | https://www.ncbi.nlm.nih.gov/geo/query/acc.cgi?acc=GSE145177 | NCBI Gene Expression Omnibus, GSE145177 |

The following previously published datasets were used:

| Author(s) | Year | Dataset title | Dataset URL | Database and Identifier |
| --- | --- | --- | --- | --- |
| Maezawa S, Barski A, Namekawa S | 2017 | RNA-seq in spermatogonia from PRC1ctrl and dKO mice | http://www.ncbi.nlm.nih.gov/geo/query/acc.cgi?acc=GSE102783 | NCBI Gene Expression Omnibus, GSE102783 |
| Kubo N, Sasaki H | 2015 | Methylome and gene expression of neonatal prospermatogonia and early postnatal undifferentiated and differentiating spermatogonia | https://ddbj.nig.ac.jp/DRASearch/submission?acc=DRA002477 | DDBJ/GenBank/EMBL, DRA002477 |
| Hobbs RM, La HM, Mäkelä J, Chan A, Rossello FJ, Nefzger CM, Legrand JM, Seram M, Polo JM | 2018 | Analysis of gene expression in populations of adult undifferentiated spermatogonia | http://www.ncbi.nlm.nih.gov/geo/query/acc.cgi?acc=GSE107124 | NCBI Gene Expression Omnibus, GSE107124 |
| Licatalosi DD, Zagore LL | 2018 | Dazl maintains proliferating germ cells through a network of polyA-proximal mRNA interactions [P6 iCLIP] | http://www.ncbi.nlm.nih.gov/geo/query/acc.cgi?acc=GSE108183 | NCBI Gene Expression Omnibus, GSE108183 |
| Naqvi S, Page DC | 2019 | Conservation, acquisition, and functional impact of sex-biased gene expression in mammalian tissues | https://www.ncbi.nlm.nih.gov/geo/query/acc.cgi?acc=GSE125483 | NCBI Gene Expression Omnibus, GSE125483 |

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

# Appendix 1

## Appendix 1—key resources table

| Reagent type (species) or resource | Designation | Source or reference | Identifiers | Additional information |
|---|---|---|---|---|
| Gene (*Mus musculus*) | deleted in azoospermia-like (*Dazl*) | Mouse Genome Informatics | MGI: 1342328 | |
| Strain, strain background (*Mus musculus*) | B6 | Taconic | TAC: B6-F TAC: B6-M | C57BL/6NTac |
| Genetic reagent (*Mus musculus*) | *Dazl:tdTomato* | *Nicholls et al., 2019b* | | B6D2-*Dazl*$^{em1(tdTomato)Huyc}$ |
| Genetic reagent (*Mus musculus*) | *Ddx4*$^{Cre}$ | *Hu et al., 2013* | MGI:5554579 | *Ddx4*$^{tm1.1(cre/mOrange)Dcp}$ |
| Genetic reagent (*Mus musculus*) | *Dazl*$^{2L}$ | *Nicholls et al., 2019b* | | B6N-*Dazl*$^{em1Dcp}$ |
| Genetic reagent (*Mus musculus*) | *Dazl*$^{-}$ | *Ruggiu et al., 1997* | RRID:IMSR_JAX:023802 | *129P2-Dazl*$^{tm1Hjc}$ |
| Genetic reagent (*Mus musculus*) | *Dazl*$^{1L}$ | *Nicholls et al., 2019b* | | B6N-*Dazl*$^{em1.1Dcp}$ |
| Genetic reagent (*Mus musculus*) | *Pou5f1:EGFP* | *Szabó et al., 2002* | RRID:IMSR_JAX:004654 | *Tg(Pou5f1-EGFP)2Mnn* |
| Genetic reagent (*Mus musculus*) | *ROSA26*$^{tdTomato}$ | *Madisen et al., 2010* | RRID:IMSR_JAX:007909 | *Gt(ROSA) 26Sor*$^{tm9(CAG-tdTomato)Hze}$ |
| Genetic reagent (*Mus musculus*) | *Rpl22*$^{HA}$ | *Sanz et al., 2009* | RRID:IMSR_JAX:011029 | RiboTag allele; B6N.129-*Rpl22*$^{tm1.1Psam}$/J |
| Antibody | anti-DAZL (rabbit polyclonal) | Abcam | Abcam ab34139; RRID:AB_731849 | IHC 1:200; iCLIP 10 μg antibody per 100 μl Dynabeads and 500 μl lysate |
| Antibody | anti-DAZL (mouse monoclonal IgG$_1$) | BioRad | BioRad MCA2336; RRID:AB_2292585 | IF 1:100 |
| Antibody | anti-FOXC2 (sheep polyclonal) | R&D Systems | R&D Systems AF6989; RRID:AB_10973139 | IF 1:250 |
| Antibody | anti-HA (mouse monoclonal IgG$_1$) | BioLegend | BioLegend #901513; previously Covance #MMS-101R; RRID:AB_2565335 | IP 2.5 μl antibody per 400 μl lysate and 200 μl Dynabeads |

*Continued on next page*

*Appendix 1—key resources table continued*

| Reagent type (species) or resource | Designation | Source or reference | Identifiers | Additional information |
|---|---|---|---|---|
| Antibody | anti-KIT (goat polyclonal) | R&D Systems | R&D Systems AF1356; RRID:AB_354750 | IF 1:250 |
| Antibody | anti-mCherry (goat polyclonal) | SICGEN | SICGEN AB0040-200; RRID:AB_2333092 | IF 1:300 |
| Antibody | anti-SOX9 (rabbit polyclona) | EMD Millipore | EMD Millipore AB5535; RRID:AB_2239761 | IF 1:300 for postnatal testis sections; 1:250 for adult testis sections |
| Antibody | anti-STRA8 (rabbit polyclonal) | Abcam | Abcam ab49405; RRID:AB_945677 | IHC 1:500 |
| Antibody | anti-ZBTB16 (goat polyclonal) | R&D Systems | R&D Systems AF2944; RRID:AB_2218943 | IF 1:250 |
| Antibody | IgG control, anti-rabbit polyclonal | Santa Cruz Biotechnologies | Santa Cruz Biotechnologies sc-2027; RRID:AB_737197 | iCLIP 10 μg antibody per 100 μl Dynabeads and 500 μl lysate |
| Commercial assay or kit | ImmPACT DAB Peroxidase Substrate | Vector Laboratories | Vector Laboratories SK-4105 | |
| Commercial assay or kit | ImmPRESS HRP anti-Rabbit Detection Kit | Vector Laboratories | Vector Laboratories MP-7401–50 | |
| Commercial assay or kit | TruSeq Stranded mRNA kit | Illumina | Illumina 20020594 | |
| Commercial assay or kit | SMART-Seq v4 Ultra Low Input RNA Kit | Takara Bio | Takara Bio 634888 | |
| Commercial assay or kit | SMARTer Stranded Total RNA-Seq Kit v2 – Pico Input | Takara Bio | Takara Bio 634411 | |
| Chemical compound, drug | N,N′-Octamethylenebis(2,2-dichloroacetamide) [Win18,446] | Santa Cruz Biotechnology | Santa Cruz Biotechnologies sc-295819 | Used at 0.1 mg/gram body weight |
| Software, algorithm | ASPeak v.2.0.0 | *Kucukural et al., 2013* | RRID:SCR_000380 | https://sourceforge.net/projects/as-peak |
| Software, algorithm | CellProfiler v3.1.8 | *Kamentsky et al., 2011* | RRID:SCR_007358 | https://cellprofiler.org |
| Software, algorithm | coRdon v1.2.0 | | | https://github.com/BioinfoHR/coRdon |
| Software, algorithm | Cutadapt v.1.8 | *Martin, 2011* | RRID:SCR_011841 | https://cutadapt.readthedocs.io/en/stable/ |
| Software, algorithm | DESeq2 v1.26.0 | *Love et al., 2014* | RRID:SCR_015687 | http://bioconductor.org/packages/release/bioc/html/DESeq2.html |
| Software, algorithm | FASTX-Toolkit v.0.0.14 | | RRID:SCR_005534 | http://hannonlab.cshl.edu/fastx_toolkit/index.html |

*Continued on next page*

*Appendix 1—key resources table continued*

| Reagent type (species) or resource | Designation | Source or reference | Identifiers | Additional information |
|---|---|---|---|---|
| Software, algorithm | HOMER v4.9.1 | *Heinz et al., 2010* | RRID:SCR_010881 | http://homer.ucsd.edu/homer/ |
| Software, algorithm | GSEA v4.0.3 | *Mootha et al., 2003; Subramanian et al., 2005* | RRID:SCR_003199 | https://www.gsea-msigdb.org/gsea/index.jsp |
| Software, algorithm | kallisto v0.44.0 | *Bray et al., 2016* | RRID:SCR_016582 | https://pachterlab.github.io/kallisto/ |
| Software, algorithm | kpLogo v1.0 | *Wu and Bartel, 2017* | | http://kplogo.wi.mit.edu |
| Software, algorithm | MEME Suite v4.11.2 | *Bailey and Elkan, 1994; McLeay and Bailey, 2010* | RRID:SCR_001783 | http://meme-suite.org/ |
| Software, algorithm | MetaPlotR | *Olarerin-George and Jaffrey, 2017* | | https://github.com/olarerin/metaPlotR |
| Software, algorithm | PANTHER v.13.1 | *Mi et al., 2017* | RRID:SCR_004869 | http://www.pantherdb.org |
| Software, algorithm | STAR v.2.5.4b | *Dobin et al., 2013* | RRID:SCR_015899 | https://github.com/alexdobin/STAR |
| Software, algorithm | TargetScanMouse v7.1 | *Agarwal et al., 2015* | RRID:SCR_010845 | http://www.targetscan.org/mmu_72/ |
| Other | Collagenase, Type I | Worthington Biochemical | Worthington Biochemical LS004196 | |
| Other | Dynabeads Protein G | Thermo Fisher Scientific | Thermo Fisher Scientific 10003D | |
| Other | GlycoBlue Coprecipitant | Thermo Fisher Scientific | Thermo Fisher Scientific AM9515 | |
| Other | Novex TBE-Urea gel, 6% | Thermo Fisher Scientific | Thermo Fisher Scientific EC6865BOX | |
| Other | Protease inhibitor, EDTA Free | MilliporeSigma | MilliporeSigma 11836170001 | |
| Other | Proteinase K, RNA grade | Thermo Fisher Scientific | Thermo Fisher Scientific 25530049 | |
| Other | RNase I | Thermo Fisher Scientific | Thermo Fisher Scientific AM2295 | For DAZL iCLIP libraries, lysates were digested with 0.005 U/µg RNase I for 3 min at 37 °C and 1100 rpm on a Thermomixer R (Eppendorf) |
| Other | TURBO DNAse | Thermo Fisher Scientific | Thermo Fisher Scientific AM2238 | |

