## [Decision Letter]

**Acceptance summary:**

Male fertility in mammals is critically dependent upon the establishment, and potentially lifelong proliferation, of a unique stem cell population – spermatogonial stem cells and the closely related cell type spermatogonia. While it is known that the DAZ family of RNA-binding proteins are required for fertility across metazoan, their precise role(s) remain ambiguous. Within this paper, the authors reveal a critical role for *Dazl* in the translation of ~2,500 protein coding genes, including several with critical roles in male fertility in their own rite. Specifically, they show that DAZL is required for the expansion and differentiation of the spermatogonial progenitor population. This work elucidates a critical element of the processes underpinning the initiation of spermatogenesis at puberty and the aetiology of male infertility.

**Decision letter after peer review:**

Thank you for submitting your article "DAZL mediates a broad translational program regulating expansion and differentiation of spermatogonial progenitors" for consideration by *eLife*. Your article has been reviewed by four peer reviewers, including Moira K O’Bryan as the Reviewing Editor and Reviewer #1, and the evaluation has been overseen by Anna Akhmanova as the Senior Editor. The following individuals involved in review of your submission have agreed to reveal their identity: Jon Oatley (Reviewer #2); Shosei Yoshida (Reviewer #4).

The reviewers have discussed the reviews with one another and the Reviewing Editor has drafted this decision to help you prepare a revised submission.

Summary:

In this study, Mikedis and colleagues explored the mechanism of action for DAZL in spermatogenesis by engineering a novel conditional knockout mouse line to refine understanding of the phenotype for a spermatogonial specific loss-of-function. They also utilized a spermatogenic synchronization strategy with RNA-seq and iCLIP technology to define the RNA binding repertoire of DAZL in mouse spermatogonia.

Overall, the manuscript is elegantly written and the data are convincing. The reviewers are in agreement that this manuscript has a strong potential to make a valuable contribution to the literature. This work is partially confirmatory of that published in Li et al., 2019, however, the reviewers see the two studies as being complementary and thus of importance in their own rite.

Essential revisions:

The only major outstanding question is which spermatogonial subsets are affected by *Dazl* deletion. Specifically, where exactly in the spectrum of spermatogonial stem cell, progenitors (early and late) to committed spermatogonia, does DAZL play key roles. For example, what is the cell cycle status of FOX2+ cells? This could be investigated using Edu incorporation assays, although we accept other experiments may be equally illuminating.

---

## [Author Response]

Essential revisions:The only major outstanding question is which spermatogonial subsets are affected by Dazl deletion. Specifically, where exactly in the spectrum of spermatogonial stem cell, progenitors (early and late) to committed spermatogonia, does DAZL play key roles. For example, what is the cell cycle status of FOX2+ cells? This could be investigated using Edu incorporation assays, although we accept other experiments may be equally illuminating.

We have modified the text to clarify that loss of *Dazl* has minimal impact on the FOXC2-positive stem/early progenitor population, which is only modestly increased in our *Dazl*cKO (Figure 1C). Instead, loss of *Dazl* reduces the size of the broader population of ZBTB16-positive undifferentiated spermatogonia (Figure 1C). These results are supported by bulk RNA-seq analysis, which shows that the expression of progenitor and undifferentiated spermatogonial markers is reduced, particularly relative to that of stem/progenitor markers (Figure 1—figure supplement 3D). Taken together, these data indicate that *Dazl* functions in progenitors to support the expansion of late progenitors. Our current results do not allow us determine whether this role in progenitor expansion is the result of DAZL activity in early vs. late progenitors.

Our data also show that loss of *Dazl* results in the formation of fewer KIT-positive differentiating spermatogonia, even after controlling for the size of the undifferentiated spermatogonial pool (Figure 1C). We conclude that DAZL functions in progenitor spermatogonia to facilitate their transition into differentiating spermatogonia.

In the Discussion, we explain that our results do not exclude a role for DAZL in spermatogonial stem cells, as recently reported by Li et al., 2019. Li et al. were able to efficiently delete *Dazl* in the spermatogonial stem cells and thereby saw a defect in this population. The delayed recombination in our model results in a different phenotype and thereby reveals a role for *Dazl* in progenitor spermatogonia.

We thank the reviewers for suggesting the EdU experiments, which we plan to include in a follow-up study.